# Objective comparison of methods to decode anomalous diffusion

Gorka Muñoz-Gil [1], Giovanni Volpe [2✉], Miguel Angel Garcia-March[3], Erez Aghion[4], Aykut Argun[2], Chang Beom Hong[5], Tom Bland[6], Stefano Bo [4], J. Alberto Conejero[3], Nicolás Firbas [3], Òscar Garibo i Orts [3], Alessia Gentili [7], Zihan Huang [8], Jae-Hyung Jeon [5], Hélène Kabbech [9], Yeongjin Kim[5], Patrycja Kowalek [10], Diego Krapf [11], Hanna Loch-Olszewska [10], Michael A. Lomholt[12], Jean-Baptiste Masson [13], Philipp G. Meyer [4], Seongyu Park [5], Borja Requena [1], Ihor Smal[9], Taegeun Song [5,14,15], Janusz Szwabiński[10], Samudrajit Thapa [16,17,18], Hippolyte Verdier [13], Giorgio Volpe [7], Artur Widera [19], Maciej Lewenstein [1,20], Ralf Metzler [16] & Carlo Manzo [1,21✉]

Deviations from Brownian motion leading to anomalous diffusion are found in transport dynamics from quantum physics to life sciences. The characterization of anomalous diffusion from the measurement of an individual trajectory is a challenging task, which traditionally relies on calculating the trajectory mean squared displacement. However, this approach breaks down for cases of practical interest, e.g., short or noisy trajectories, heterogeneous behaviour, or non-ergodic processes. Recently, several new approaches have been proposed, mostly building on the ongoing machine-learning revolution. To perform an objective comparison of methods, we gathered the community and organized an open competition, the Anomalous Diffusion challenge (AnDi). Participating teams applied their algorithms to a commonly-defined dataset including diverse conditions. Although no single method performed best across all scenarios, machine-learning-based approaches achieved superior performance for all tasks. The discussion of the challenge results provides practical advice for users and a benchmark for developers.

[1] ICFO – Institut de Ciències Fotòniques, The Barcelona Institute of Science and Technology, Av. Carl Friedrich Gauss 3, 08860 Castelldefels (Barcelona), Spain. [2] Department of Physics, University of Gothenburg, Origovägen 6B, SE-41296 Gothenburg, Sweden. [3] Instituto Universitario de Matemática Pura y Aplicada, Universitat Politècnica de València, Valencia, Spain. [4] Max Planck Institute for the Physics of Complex Systems, Nöthnitzer Straße 38, DE-01187 Dresden, Germany. [5] Department of Physics, Pohang University of Science and Technology, Pohang 37673, Korea. [6] The Francis Crick Institute, 1 Midland Road, London NW1 1AT, UK. [7] Department of Chemistry, University College London, 20 Gordon Street, London WC1H 0AJ, UK. [8] School of Physics and Electronics, Hunan University, Changsha 410082, China. [9] Department of Cell Biology, Erasmus University Medical Center, Dr. Molewaterplein 40, 3015 GD Rotterdam, the Netherlands. [10] Faculty of Pure and Applied Mathematics, Hugo Steinhaus Center, Wrocław University of Science and Technology, Wrocław, Poland. [11] Department of Electrical and Computer Engineering, Colorado State University, Fort Collins, Colorado 80523, USA. [12] PhyLife, Department of Physics, Chemistry and Pharmacy, University of Southern Denmark, DK-5230 Odense M, Denmark. [13] Institut Pasteur, Université de Paris, USR 3756 (C3BI/DBC) & Neuroscience department CNRS UMR 3751, Decision and Bayesian Computation lab, F-75015 Paris, France. [14] Center for AI and Natural Sciences, Korea Institute for Advanced Study, Seoul, Korea. [15] Department of Data Information and Physics, Kongju National University, Kongju 32588, Korea. [16] Institute of Physics & Astronomy, University of Potsdam, Karl-Liebknecht-Str 24/25, D-14476 Potsdam-Golm, Germany. [17] Sackler Center for Computational Molecular and Materials Science, Tel Aviv University, Tel Aviv 69978, Israel. [18] School of Mechanical Engineering, Tel Aviv University, Tel Aviv 69978, Israel. [19] Department of Physics and Research Center OPTIMAS, Technische Universität Kaiserslautern, 67663 Kaiserslautern, Germany. [20] ICREA, Pg. Lluís Companys 23, 08010 Barcelona, Spain. [21] Facultat de Ciències i Tecnologia, Universitat de Vic – Universitat Central de Catalunya (UVic-UCC), C. de la Laura,13, 08500 Vic, Spain. ✉email: giovanni.volpe@physics.gu.se; carlo.manzo@uvic.cat

The random walk[1] is a mathematical model ubiquitously employed at all scales in a variety of scientific fields, including physics, chemistry, biology, ecology, psychology, economics, sociology, and computer science (Fig. 1a)[2,3]. Random walks are characterized by an erratic change of an observable over time (e.g., position, temperature, or stock price, Fig. 1b). The archetypal example of a random walk is Brownian motion, which describes the movement of a microscopic particle in a fluid as a consequence of thermal forces[4].

The space explored by random walkers over time is commonly measured by the mean squared displacement (MSD), which grows linearly in time for Brownian walkers (MSD $\propto t$)[4]. Deviations from such a linear behavior displaying an asymptotic power-law dependence (MSD $\propto t^\alpha$) have been observed in several fields and are generally referred to as anomalous diffusion[4]: subdiffusion for $0 < \alpha < 1$, and superdiffusion for $\alpha > 1$ (as particular cases, $\alpha = 0$ corresponds to immobile trajectories, $\alpha = 1$ to Brownian motion, and $\alpha = 2$ to ballistic motion). The left panel in Fig. 1c shows some examples of MSDs for Brownian (black line), subdiffusive (blue line), and superdiffusive (red line) motion together with the corresponding trajectories in 2D. For example, anomalous diffusion occurs in the diffusion of lipids and receptors in the cell membrane[5], in the transport of molecules within the cytosol[6] and the nucleus[7], in the foraging and mating strategies of animals[8], in sleep-wake transitions during sleep[9], and in the fluctuations of the stock market[10].

The recurrent observation of anomalous diffusion has driven an important theoretical effort to understand and mathematically describe its underlying mechanisms. This effort has provided a palette of microscopic models characterized by different spatial (step length) and temporal (step duration) random distributions, both with and without long-range correlations[4]. Important models for the interpretation of experimental results are continuous-time random walk (CTRW)[11], fractional Brownian motion (FBM)[12], Lévy walk (LW)[13], annealed transient time motion (ATTM)[14], and scaled Brownian motion (SBM)[15] (some sample trajectories are shown in the central panel of Fig. 1c, see Methods, "Theoretical models").

In typical experiments aimed at understanding diffusion, the available data consists of trajectories of a tracer, such as a molecule in a cell, a stock price in the stock market, a foraging animal in its environment. The aim is to extract from these trajectories information about properties of the tracer and of the medium where its motion takes place, namely to infer the anomalous diffusion exponent $\alpha$, to determine the underlying diffusion model and, finally, to determine whether these properties change over time and space.

The first crucial step to characterize the tracer's motion is the determination of the anomalous diffusion exponent $\alpha$

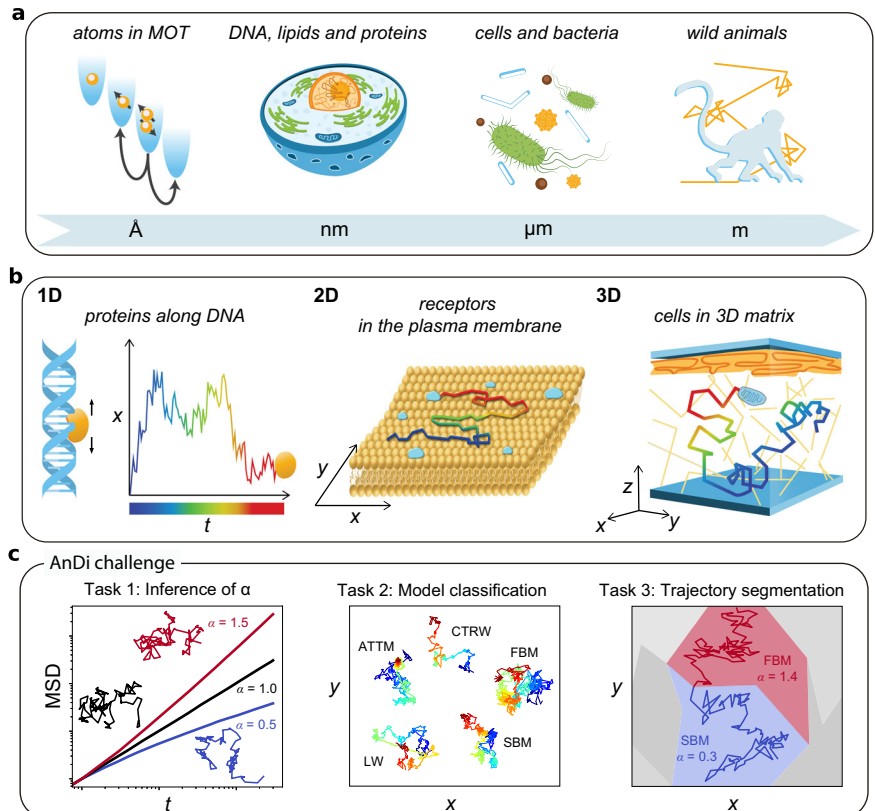

**Fig. 1 The AnDi challenge tasks and datasets. a** Random walks, characterized by an erratic change of an observable, occur at all length and time scales in a variety of systems. Examples are provided by atoms in magneto-optical traps; the diffusion of cellular components, such as DNA, proteins, lipids, and organelles; the motion of bacteria and cells; and animals foraging and mating. **b** Trajectories of tracers in spaces of different dimensionality: 1D, Proteins sliding along DNA fragments; 2D, receptors diffusing in the plasma membrane; 3D, cells migrating in a 3-dimensional matrix. The color code of the trajectories represents time. **c** The challenge tasks. Task 1 – Inference of the anomalous diffusion exponent. Representative trajectories and corresponding MSD for diffusive ($\alpha = 1$, black lines), subdiffusive ($0 < \alpha < 1$, blue lines), and superdiffusive ($1 < \alpha < 2$, red lines) motion. Task 2 – Classification of the underlying anomalous diffusion model. Representative trajectories for a continuous-time random walk (CTRW), fractional Brownian motion (FBM), Lévy walk (LW), annealed transient time motion (ATTM), and scaled Brownian motion (SBM). Different diffusion models produce subtle changes. Details of the models are described in the text and in Methods, "Theoretical models". Task 3 – Segmentation and characterization of a trajectory with changepoint. Trajectory switching diffusion model and/or exponent as a result of diffusion in a spatially heterogeneous environment, represented by the colored patches.

(Task 1, Fig. 1c). It is typically estimated by fitting the MSD to a power law[16]. Traditionally, the MSD is defined as the ensemble average over a group of tracers (EA-MSD, Equation (1)), in analogy to the solution to Fick's second law for the spreading of a bunch of particles in a homogeneous medium[4]. When long tracks are available, the MSD can be instead obtained as a time average from the trajectory of a single tracer (TA-MSD, Equation (2)). While seemingly a straightforward procedure, determining $\alpha$ from the MSD can introduce significant errors and biases: (i) the accuracy of the estimation depends on fluctuations, which can only be reduced by increasing the number of tracers (for EA-MSD) or the length of the trajectory (for TA-MSD), which is often not possible because of practical constraints; (ii) the value of $\alpha$ is biased by noise, such as the localization precision of experimental trajectories[17], which needs to be estimated independently to introduce a proper correction[16,18]; iii) while for a stationary motion in a homogeneous medium, EA-MSD and TA-MSD have the same exponent, several systems are intrinsically heterogeneous and non-stationary[19,20], which can lead to non-ergodicity (i.e., the non-equivalence of time and ensemble averages). Typically, the exponent $\alpha$ of the EA-MSD characterizes the physical properties of the systems (e.g., the trapping time distribution in CTRW or the time-dependence of diffusivity in SBM). However, in several non-ergodic systems, the TA-MSD shows a linear behavior with respect to the timelag in the long time limit even when $\alpha \neq 1$[4]; iv) the behavior of the MSD at short times or timelags might differ from its asymptotic limit[4], thus long trajectories are required for the correct estimation of $\alpha$.

The second critical issue is to determine the underlying diffusion model (Task 2, Fig. 1c), which is related to its driving physical mechanism. Here, difficulties arise because the calculation of the MSD is not very informative, since different models provide curves with the same scaling exponent. Other statistical parameters have been proposed for this task and algorithms based on the combination of several estimators allow to distinguish between pairs of models[21–24], but there is no general consensus on how to unambiguously determine the underlying diffusion model from a trajectory.

The third issue is to determine whether the properties of the motion of a given tracer change over time[6,20,25,26] (Task 3, Fig. 1c). This can be both the result of heterogeneity in the environment (e.g., patches with different viscosity on a cellular membrane) or of time-varying properties of the tracer (e.g., different activation states of a molecular motor). In these cases, the determination of $\alpha$ and of the underlying diffusion model must be combined with a segmentation of the trajectory to identify fragments with homogeneous characteristics. Several methods have been proposed for the segmentation of time traces[27], mostly based on changes in diffusion constant, velocity, or diffusion mode (e.g., immobile, random, directed)[28–31]. Only recently, attempts have been made to determine changepoints with respect to a switch in $\alpha$[25,32,33] and diffusion model[34]. Until now, a systematic assessment of changepoint detection methods for anomalous diffusion has not been performed.

In recent years, advances in fluorescence techniques have greatly increased the availability of high-precision trajectories of single molecules in living systems[35], producing an increasing drive to develop methods for quantifying anomalous diffusion[16,25,32,36–39]. Furthermore, the recent blossoming of machine learning has promoted the accessibility of new powerful tools for data analysis[40] and further widened the palette of available methods[33,41–43]. Some of the novel approaches have already delivered new insights into anomalous diffusion in different scenarios[44–46].

This recent increase of available methods performing similar tasks requires an objective assessment on a common reference dataset to define the state of the art and guide end-users in the optimal choice of characterization tools for each specific application. To assess the performance in quantifying anomalous diffusion, we have therefore run an open competition, the Anomalous Diffusion (AnDi) Challenge, divided into three different tasks: anomalous exponent inference, model classification, and trajectory segmentation, each for 1D, 2D, and 3D trajectories. The performance of submitted methods was assessed with common metrics on simulated datasets with trajectory length and signal-to-noise level reproducing realistic experimental conditions (Methods, "Structure of the datasets"). The submitted methods were also compared on the blind analysis of experimental trajectories (Supplementary Note 2). Although several experiments provide 2D and 3D trajectories, we first present and discuss in detail the results obtained for the 1D trajectories. This choice is driven by the fact that the 1D-case is conceptually easier to understand, thus complex methods are in general first developed in 1D and then extended to multidimensional space, as testified by the larger participation for this dimension. Thus, it allows us to assess the performance of a larger set of methods including those that might eventually be extended to 2D and 3D. We follow the same rationale when describing the physical models and their simulations.

## Results

**Competition design**. The challenge consisted of three tasks: Task 1 (T1) – inference of the anomalous diffusion exponent $\alpha$; Task 2 (T2) – classification of the underlying diffusion model; Task 3 (T3) – trajectory segmentation (Fig. 1c and Methods, "Organization of the challenge"). The aim of the last task was to identify the changepoint within a trajectory switching $\alpha$ and diffusion model, as well as to determine the exponent and model for the identified segments. Each task was further divided into three subtasks corresponding to the trajectory dimensions (1D, 2D, and 3D, Fig. 1b), totaling 9 independent subtasks. Participants could choose to submit predictions for any combination of subtasks. For the competition, we let developers build and use their own tools to provide predictions for the common dataset. While this choice limited the methods assessed to those provided by the community, it ensured that those algorithms were properly applied. Datasets were generated as described in Methods, "Structure of the datasets" and "Theoretical models".

**Challenge participants and performance evaluation**. We received submissions from 13 teams for T1, 14 teams for T2, and 4 teams for T3. One of the methods participating in T3 had results comparable with random predictions and was thus excluded from the discussion of the results. Basic information about methods used by participating teams can be found in Methods, "Challenge methods", Table 1, and Supplementary Note 1. A detailed description of each of the methods can be found in the referenced articles.

We investigated the performance of the methods submitted for each task separately using the metrics described in Methods, "Metrics". A summary of rankings for all tasks and methods is presented in Supplementary Fig. 2. Full rankings for T1 and T2 in all dimensions are presented in Fig. 2a–c and Fig. 3a–c, respectively, together with representative information for the best-in-class methods for the 1D case (Fig. 2d–g and Fig. 3d–g, respectively). The same analysis is presented in Supplementary Fig. 3 and Supplementary Fig. 4 for higher dimensions. Results for T3 in 1D are shown in Fig. 4a–c, together with representative information for the best-in-class methods (Fig. 4e, f). Results for all dimensions are presented in Figs. 4d, e and Supplementary Fig. 5.

**Task 1: Inference of the anomalous diffusion exponent**. The inference of the exponent $\alpha$ is the most popular way to quantify anomalous diffusion and 13 teams participated in T1 of the AnDi

**Table 1 Participating teams and summary of methods. See Supplementary Note 1 for further details on these methods. Methods were classified based on the type of approach (as machine learning (ML), or classical statistics (Stat)); their input data (as raw/ lightly preprocessed trajectories (Traj), or features (Feat)); and their training procedure (as length-specific (L-specific, Yes), or not (No)).**

| Label | Team name | Method | Class | Input | Tasks | L-specific |
|---|---|---|---|---|---|---|
| A | Anomalous Unicorns | Ensemble of CNN and RNN[45,76] | ML | Traj | T1(1D), T2(1D) | No |
| B | BIT | Bayesian inference[77,78] | Stat | Traj | All | No |
| C | DecBayComp | Graph neural networks[79] | ML | Traj + Feat | T1, T2(1D, 2D) | No |
| D | DeepSPT | ResNet + XGBoost[80,81] | ML | Traj + Feat | T1(1D), T2(1D) | No |
| E | eduN | RNN + Dense NN[82] | ML | Traj | All | Yes |
| F | Erasmus MC | bi-LSTM + Dense NN[31] | ML | Feat | T1, T2 | Yes |
| G | HNU | LSTM[83] | ML | Traj | T1 | Yes |
| H | NOA | CNN + bi-LSTM[84] | ML | Traj | T1(1D) | No |
| I | QUBI | ELM[85] | ML | Feat | T1(1D), T2(1D) | No |
| J | FCI | CNN[42,86] | ML | Traj | T1(1D, 2D), T2(1D, 2D), T3(1D, 2D) | No |
| K | TSA | Scaling analysis and feature engineering[87] | Stat | Feat | T1, T2(1D) | No |
| L | UCL | Feature engineering + NN[88] | ML | Feat | T1, T2 | No |
| M | UPV-MAT | CNN + bi-LSTM[89] | ML | Traj | T1, T2 | Yes |
| N | Wust ML A | 1D: RISE + forest classifier[90] 2D, 3D: MrSEQL + logistic reg.[91] | ML | Feat | T2 | No |
| O | Wust ML B | Gradient boosting regression + classifier[43,92,93] | ML | Feat | T1(1D, 2D), T2 | No |

Challenge (Fig. 2a–c). We observed a rather large spread of performances, but for each dimension we could identify a cluster of four top methods with comparable performance, scoring better than the rest. Three methods (E, G, and L) were consistently part of the top group in all dimensions. All top teams used machine-learning approaches: teams E, G, J, and M applied them to raw or simply pre-processed trajectories; teams F and L used statistical features as inputs. All these methods, except L and J, were based on length-specific training.

Besides the overall MAE, Fig. 2a–c also shows the performance obtained for specific diffusion models (colors within bars) by all participating teams. In Fig. 2d–g, the methods are compared with the simple fitting of the TA-MSD, used as a baseline method (Methods, "Alternative and baseline estimators"). Most methods perform better than TA-MSD. As expected, the fit of the TA-MSD shows better performance on ergodic (FBM) and ultra-weakly non-ergodic (LW) rather than on (weakly) non-ergodic models (CTRW, ATTM, and SBM), for which TA-MSD and EA-MSD have different scaling exponents (Fig. 2d and Supplementary Fig. 6). Interestingly, the top-performing methods do not suffer from this limitation and provide similar MAE for all the models, with exception of the ATTM (short ATTM trajectories might not undergo any change of diffusion coefficient and, therefore, the result is indistinguishable from pure Brownian motion, impacting the final performance). As an example, in Fig. 2e, we show a 2D histogram of the predicted exponent vs the ground truth for the best-in-class method (team M) and the TA-MSD (upper inset) in 1D. As most of the top-scoring methods (Supplementary Fig. 7, Supplementary Fig. 8, and Supplementary Fig. 9), the best-in-class method achieves similar performance over the whole range of $\alpha$, whereas TA-MSD has a lower accuracy for $\alpha \simeq 0.5$ to 1. Obtaining precise predictions for $\alpha \simeq 1$ is particularly relevant, since the correct assessment of the exponent in this regime would further allow the discrimination between normal and anomalous diffusion. In addition, the method of team M (similarly to other top methods, (Supplementary Fig. 14, Supplementary Fig. 15, and Supplementary Fig. 16)) does not show any bias, whereas the TA-MSD systematically underestimates the value of $\alpha$ as a consequence of localization error[16,18] (Fig. 2e, lower inset).

In Fig. 2f, we explore the effect of the trajectory length on the exponent prediction. As the trajectory length increases, the MAE

rapidly decreases toward a value $\approx 0.1$ for the best-performing methods. Thus, the MAE of machine-learning approaches features a striking improvement with respect to the nearly constant MAE of the TA-MSD, demonstrating the capability of machine learning to take advantage of the information contained in longer trajectories.

Last, we investigate the effect of the level of noise (Fig. 2g). Even for SNR = 1, i.e., when the standard deviation of the noise has the same amplitude as the displacement standard deviation, the top-performing methods show a greater than 2-fold improvement in predicting $\alpha$ with respect to TA-MSD. Thus, while localization noise delays convergence of TA-MSD to its asymptotic behavior[16], the top methods seem able to determine patterns associated to the correct exponent even from short-time behaviors, which is an ability particularly useful for many potential applications to the analysis of experimental data.

**Task 2: Classification of the underlying diffusion model**. We present the performance of the submitted methods to classify trajectories between the 5 diffusion models in Fig. 3 and Supplementary Fig. 4. For each dimension of this task, a different number of methods showed comparable performance (Fig. 3a–c). For each dimension, we selected the 2 teams that achieved top scores. These top positions were occupied by three teams with machine-learning methods operating on raw trajectories (teams E and M) or features (team L). In general, the use of features as input to machine learning models seems to provide better results as the trajectory dimension increases.

We also dissect the results as a function of the exponent $\alpha$, as shown in Fig. 3a–c (colors within the bars), and in more detail in Fig. 3d for 1D, and in Supplementary Fig. 10 for all dimensions. For all methods, the worst performance is achieved for $\alpha \simeq 1$. This is expected because in this regime all models converge to pure Brownian motion and thus feature large similarities in their long-time statistical properties, even though their microscopic generative dynamics are different. A similar situation occurs for $\alpha \rightarrow 0$, a regime in which, independently of the underlying model, trajectories are nearly immobile and dominated by localization noise. Still, most of the methods show good predictive capability ($F_1 \gtrsim 0.7$) even in these two regimes, since they probably learn to recognize details or patterns of the microscopic dynamics. The

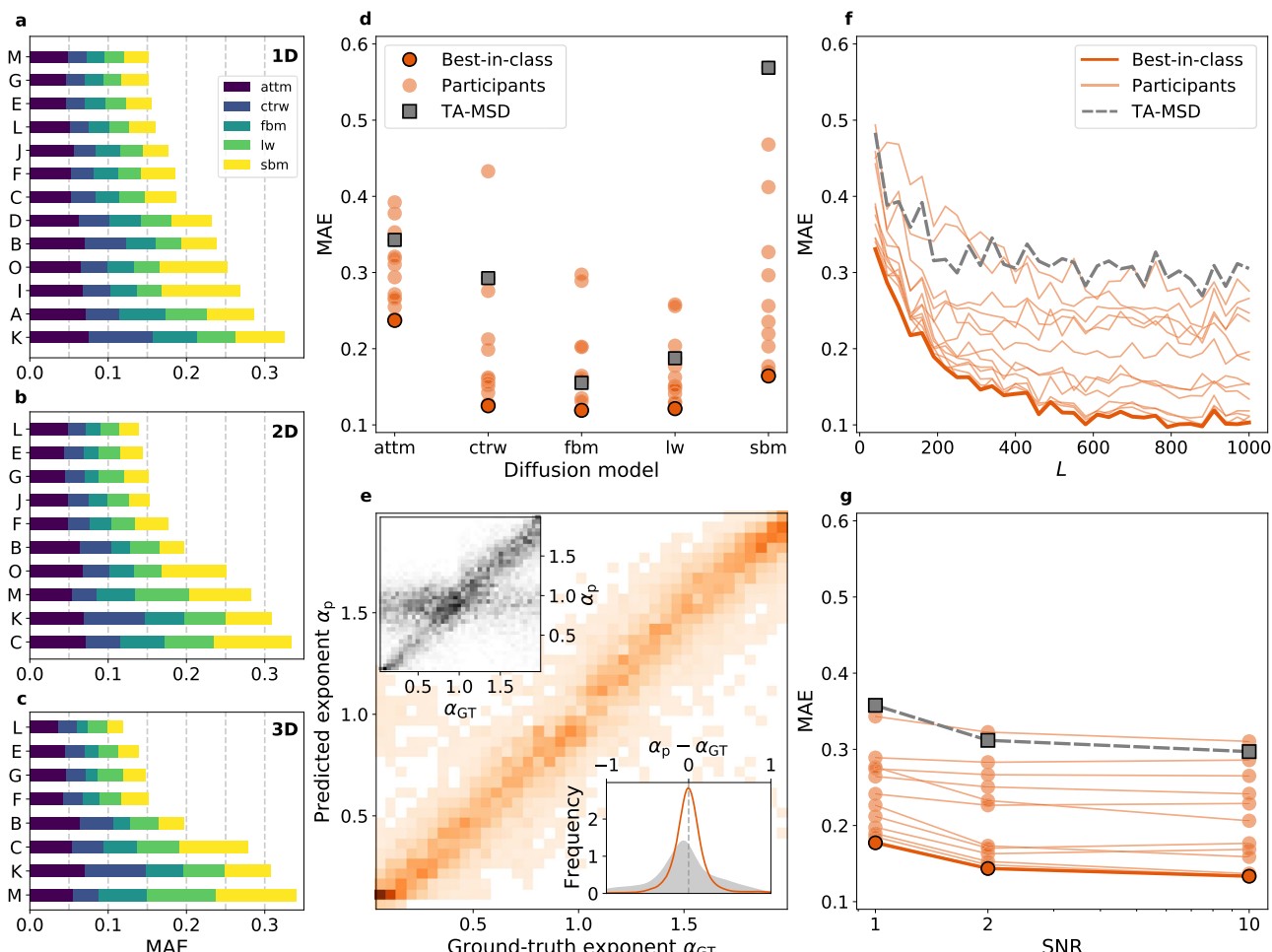

**Fig. 2 Challenge results for Task 1: inference of $\alpha$. a–c** Final leaderboards according to the MAE obtained by participants for 1D (**a**), 2D (**b**), and 3D (**c**). The colors represent the relative contributions to the overall mean absolute error (MAE) calculated for each underlying diffusion model and normalized such that the sum of all contributions gives the value of the same metric calculated over the whole dataset. **d** MAE obtained by participating teams as a function of the diffusion model for 1D trajectories. **e** Probability distribution of the predicted vs ground-truth anomalous diffusion exponent for the best-in-class team in 1D (team M). Insets: (top left) Probability distribution of the predicted vs. ground-truth anomalous diffusion exponent for the baseline method (TA - MSD). (bottom right) Frequency of the bias between predicted and ground-truth anomalous diffusion exponent for the best-in-class team (team M, orange line) and the baseline method (TA - MSD, gray area) in 1D. **f** MAE obtained by participating teams as a function of the trajectory length in 1D. **g** MAE obtained by participating teams as a function of the SNR in 1D. All results for T1 in 1D, 2D, and 3D are provided in Supplementary Figs. 3, 6-9, 14-16.

confusion matrix of the best-in-class method (team E) for the 1D subtask (Fig. 3e) provides a representative view of the classification capabilities of these methods. Results obtained by other methods are shown in Supplementary Fig. 11, Supplementary Fig. 12, and Supplementary Fig. 13. The best accuracy is obtained for CTRW and LW, for which the method of team E is able to identify their markedly different features. However, it shows a higher level of error when discriminating between Gaussian processes, such as FBM and SBM[39]. The worst performance is obtained for ATTM, whose trajectories display a large heterogeneity in diffusion coefficients and lack a characteristic time-scale. Rather long trajectories (including at least a switch of diffusivity) are thus necessary to distinguish ATTM from the other models.

Similarly to what we observe for T1, the trajectory length and the presence of localization noise affect the performance of the methods, as shown in Fig. 3f, g, respectively. Nevertheless, even for very short and noisy trajectories, the results obtained by the top methods display excellent accuracy ($F_1 \approx 0.6$ to 0.8), taking into account that the largest noise level severely hides the actual diffusive dynamics.

**Task 3: Segmentation of the trajectory**. Recently, several experimental studies have evidenced the occurrence of switching of diffusion model and $\alpha$ within individual trajectories[6,25]. However, methods to determine and analyze such changes are not established and are widely employed yet. Probably, for this reason, the participation to T3 was reduced as compared to T1 and T2, with two teams proposing machine-learning methods (RNNs for team E and CNN for team J), and team B using Bayesian inference. The methods taking part in T3 were specifically designed for the challenge and have not been tested on other time-dependent processes, e.g., such as those involving a continuous change of anomalous diffusion properties.

The main objective of T3 is the precise assessment of the changepoint between two diffusive regimes, characterized by different diffusion models and anomalous diffusion exponents. As shown in Fig. 4a, participants to this task achieved RMSE well below the one obtained from random predictions. The RMSE is heavily affected by the position of the changepoint, being minimum for changepoints located near the center of the trajectory. As described earlier, the performance for predictions of $\alpha$ and the diffusion model strongly depends on the trajectory

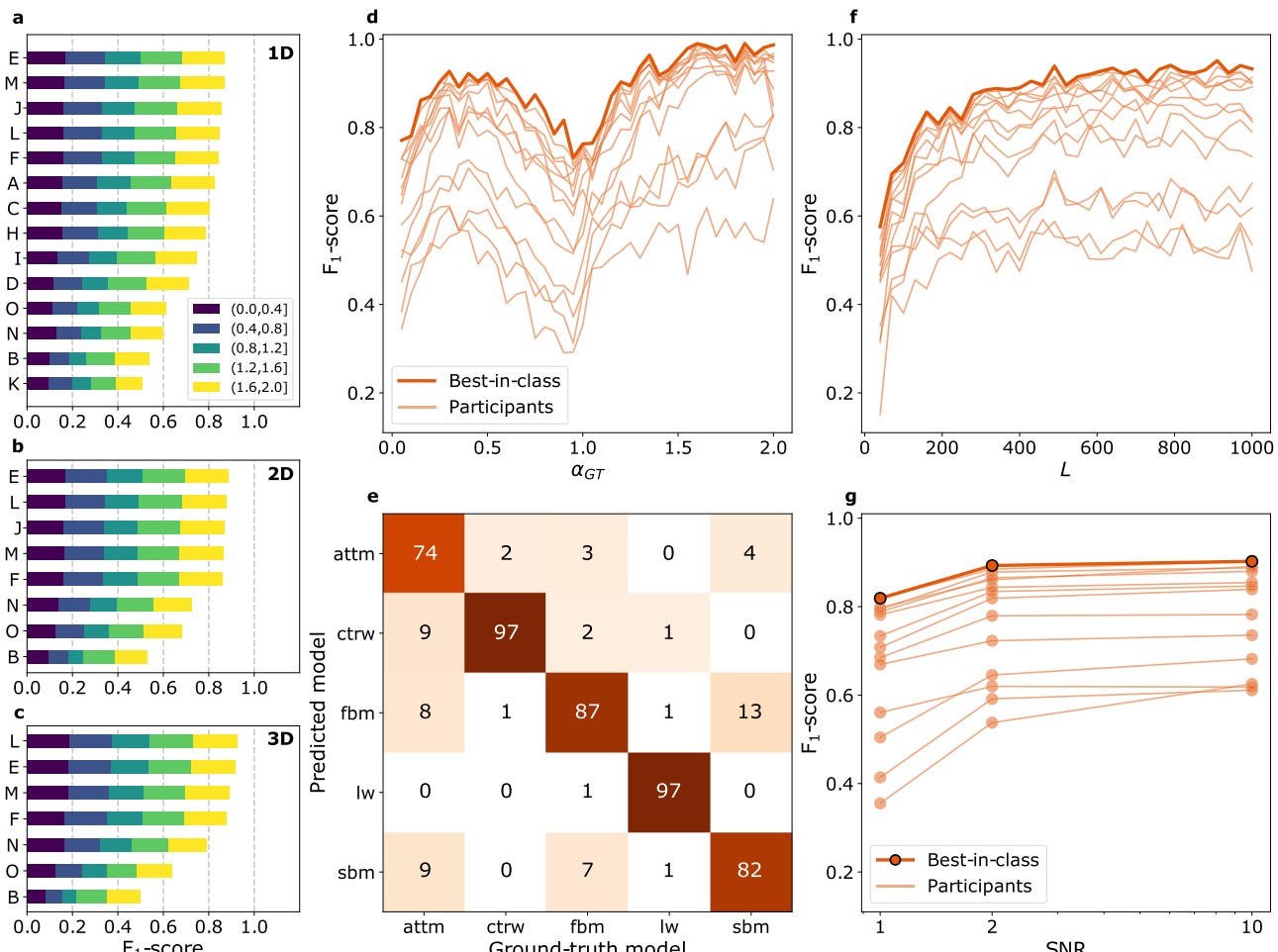

**Fig. 3 Challenge results for Task 2: diffusion model classification. a–c**, Final leaderboards according to the F$_1$-score obtained by participants for 1D (**a**), 2D (**b**), and 3D (**c**). The colors represent the relative contributions to the overall F$_1$-score calculated for different ranges of anomalous diffusion exponents and normalized such that the sum of all contributions gives the value of the same metric calculated over the whole dataset. **d** F$_1$-score obtained by participating teams as a function of the anomalous diffusion exponent for 1D trajectories. **e** Confusion matrix for the predictions of the best-in-class team in 1D (team E). Numbers in matrix cells represent the number of correctly and incorrectly classified trajectories for each ground-truth model as percentages of the number of trajectories of the corresponding ground-truth model (column-based normalization, so that their sum along the columns should add up to 100, with minor deviation due to rounding). Thus, the percentages of correctly classified observations can be thought of as class-wise recalls. **f** F$_1$-score obtained by participating teams as a function of the trajectory length in 1D. **g** F$_1$-score obtained by participating teams as a function of the SNR in 1D. All results for T2 in 1D, 2D, and 3D are provided in Supplementary Figs. 4, 10-13, 17-20.

length. In this task, they are thus correlated to the changepoint position, which sets the segment length. Therefore, the larger (smaller) the distance of the changepoint from the origin, the better (worse) the prediction for the first segment is and the worse (better) than for the second segment (Fig. 4b, c).

For the challenge purposes, we simulated all trajectories as having a changepoint that could be located at any position, including the endpoints. In this view, the presence of a changepoint at one extreme was interpreted as a trajectory not having an "actual" changepoint. Similarly, participants were required to always provide a prediction for the changepoint position. In the case of not detecting a changepoint, the predicted position should have coincided either with the start or the end point of the trajectory, considered equivalent for this evaluation. With this design, the RMSE simultaneously provides an evaluation of the localization precision as well as of its specificity. We also performed further analyses to independently assess the sensitivity and specificity of the participating methods and gain further insight into their performance. Since Fig. 4a–c show that it is challenging to estimate the changepoint when it is located very close to the trajectory start/end

points, we considered trajectories with a changepoint within $\epsilon = 20$ points from the start/end as not having a changepoint. The same criterion was applied to the predictions provided by each method. Predictions/ground truth pairs located at $\epsilon < t < L - \epsilon$ were counted as true positives. Predictions/ground truth pairs located at $t \leq \epsilon$ or $t \geq L - \epsilon$ were counted as true negatives. Mixed cases were considered as false positive or false negatives. Based on this classification, we could evaluate the recall (Equation (14)), the false positive rate (Equation (15)), and the Jaccard similarity coefficient (Equation (16)). We also calculated the RMSE$_{\text{TP}}$, defined as the RMSE obtained only for true positives.

The plot of the recall vs. the false positive rate (Fig. 4d) shows that all submitted methods detect more than 92% of the inner changepoints but present a rate of false positives larger than $\approx$10% and sometimes as high as $\approx$40%. We think that several factors might interplay to produce this behavior. As explained earlier, participants always provided a prediction for the changepoint position, the latter being equal to one of the trajectory endpoints if no changepoint was detected. In the latter analysis, our distance-based criterion relaxes this requirement to a distance $\epsilon = 20$ points from the endpoints.

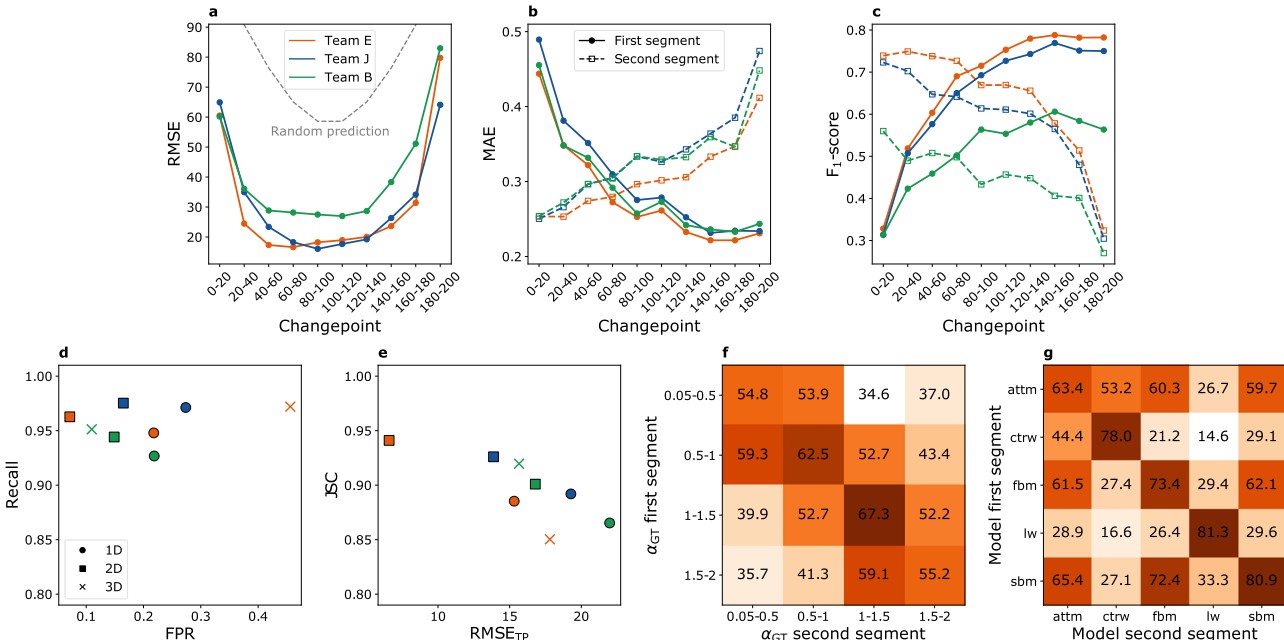

**Fig. 4 Challenge results for Task 3: trajectory segmentation and characterization. a** Root mean square error (RMSE) as a function of the changepoint location along the trajectory for all teams in 1D (teams E, J, and B). Dashed lines represents the RMSE associated to a random prediction of the changepoint position. **b** Corresponding means absolute error (MAE) of the prediction of $\alpha$ and, (**c**) $F_1$-score for the classification of the diffusion model for the first (solid symbols/continuous line) and second segment (empty symbols/dashed line) as a function of the changepoint location along the trajectory. **d** Plot of the recall vs the FPR for all participating teams. **e** Plot of JSC vs RMSE$_{TP}$ for all participating teams. For the calculation of the metrics in **d-e**, only trajectories presenting a changepoint at a distance larger than 20 points from the start/end points were considered as undergoing a switch. RMSE$_{TP}$ was estimated only for true positive position pairs. Colors indicate teams, following the same color code as in **a**. **f**, **g** RMSE as a function of the anomalous diffusion exponent (**f**) and of the diffusion model (**g**) of the first and second segment for the best-in-class team in 1D (team E). All results for T3 in 1D, 2D, and 3D are provided in Supplementary Fig. 5.

Thus, the high false-positive rate reflects the methods' limitations when dealing with changepoints close to the trajectory endpoints that, instead of being associated with no changepoint, are generally predicted to be more internal. Nevertheless, since the challenge metric does not explicitly account for false-positive identifications, predicting an inner changepoint even when the odds of predicting a false positive are high might be a conservative choice to keep the RMSE low. In some cases, this effect is produced by the choice of architecture. For example, in 1D and 3D, team E applied a strategy based on the averaging of predictions obtained through different networks. In this way, they could reduce the RMSE even for changepoints close to the trajectory endpoints (Fig. 4a), but it also led to a high rate of false positives (Fig. 4d, e), associated with contrasting predictions of the networks (e.g., a very early changepoint and a very late changepoint), averaging into an internal point.

In addition, we aimed at exploring the relationship between the overall detection performance and changepoint localization precision. As a measure of detection performance, we used the Jaccard similarity coefficient for binary classification (Equation (16)) that, with respect to the recall, further accounts for false-positive detection. The localization precision was instead estimated by RMSE$_{TP}$ resulting from true positive identifications. The plot of the Jaccard similarity coefficient vs RMSE$_{TP}$ (Fig. 4e) shows that, despite the false positive rate, all submitted methods show good overall detection performance and comparable precision (RMSE$_{TP}$ = 10–20 points). Interestingly, the performance of teams B and J improves with the dimensionality of the problem, consistently with the increase of information provided by the additional components of the motion. Team E also shows an improvement from 1D to 2D, in agreement with this explanation. The degradation of performance of team E in 3D

can be ascribed to their approach to the problem through the independent training of three 1D networks, showing obvious limitations when applied to a diffusion model that is not the simple composition of 1D diffusion along with orthogonal directions.

The combination of $\alpha$-exponents and diffusion models of the two segments is also expected to affect the changepoint localization precision. However, our dataset has a rich parameter space entangling several variables (anomalous model, $\alpha$, noise, changepoint location) and some imbalance since not all the models can have any value of $\alpha$. To highlight changes in RMSE due to a switch in $\alpha$ or in the diffusion model, we restricted the analysis to a subset of trajectories with a single noise level (SNR=10, Fig. 4f, g). Unsurprisingly, the RMSE is minimal when there is a large change in $\alpha$, as between nearly immobile motion ($\alpha < 0.5$) to either superdiffusion ($1 \leq \alpha < 1.5$) or directed or ballistic motion ($1.5 \leq \alpha \leq 2$) (Fig. 4f). The worst-case scenario is instead observed when both segments undergo mild sub- ($0.5 \leq \alpha < 1$) or superdiffusion ($1 \leq \alpha < 1.5$). The matrix shows a reasonable level of symmetry, considering the large heterogeneity of the dataset. However, in the presence of small changes of $\alpha$, such as between $0.05 \leq \alpha < 0.5$ and $0.5 \leq \alpha < 1$, or between $1 \leq \alpha < 1.5$ and $1.5 \leq \alpha \leq 2$, the methods seem to detect changes involving an increase of $\alpha$ with better precision.

This dependence is related in a nontrivial fashion to the change in RMSE observed as a function of diffusion models (Fig. 4g). In fact, while FBM and SBM allow Brownian, sub-, and superdiffusion, CTRW and ATTM do not allow superdiffusion, and LW does not allow subdiffusion. Changepoints associated with a switch of $\alpha$ but with no change of model are the most difficult to precisely locate. The smallest RMSE is observed when LW switches to CTRW. In contrast, models involving an abrupt

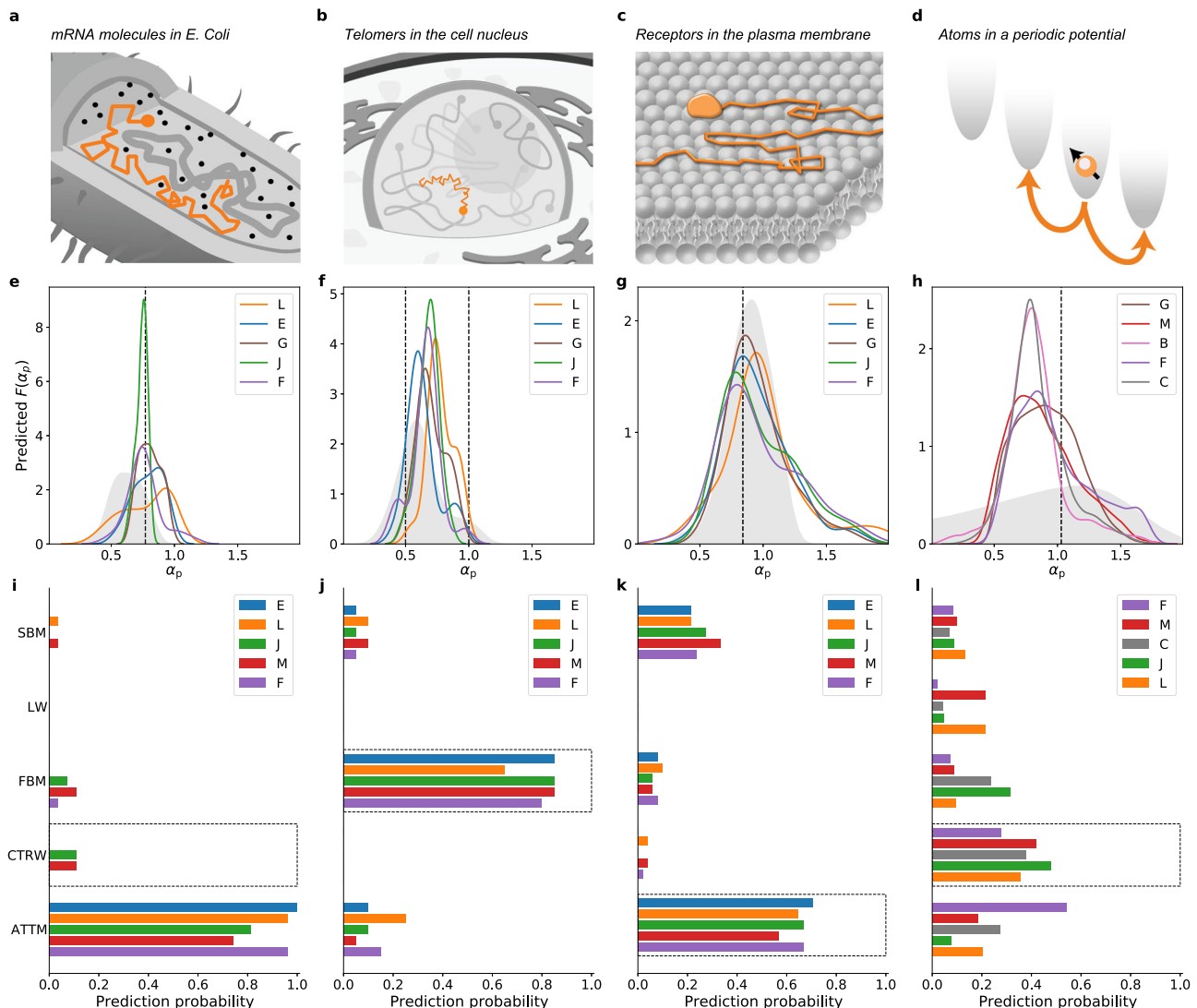

**Fig. 5 Analysis of experimental datasets. a–d,** Schematic representation of the experiments analyzed in the contest of the AnDi challenge: 2D motion of mRNA molecules inside live E. coli cells ([47], **a**); 2D motion of telomeres in the nucleus of mammalian cells ([38,48], **b**); 2D motion of biomolecular receptors moving on the membrane of mammalian cells ([20], **c**); and 1D motion of single atoms moving in a 1D periodic optical potential ([49], **d**). **e–h** Histograms of the estimation of the anomalous diffusion exponent $\alpha_p$ predicted by top teams for trajectories from experimental datasets. Gray areas correspond to the results of the baseline method TA-MSD. Dashed lines indicate the original estimations of $\alpha$ provided by Refs. [47] (**e**),[38,48] (**f**),[20] (**g**), and[49] (**h**). **i–l** Histograms of the diffusion model predicted by top teams for trajectories from experimental datasets. Dashed boxes indicate the original classifications provided by Refs. [47] (**i**),[38,48] (**j**),[20] (**k**), and[49] (**l**). We show predictions obtained by the top 5 teams for the corresponding subtask. For the last dataset, we further selected the teams based on their performance on short ($L \approx 10$) trajectories. All results for the analysis of the experimental data are presented in Supplementary Figs. 21-28.

(ATTM) or smooth change of diffusivity (SBM) are the most difficult to distinguish from the others.

**Analysis of experimental data.** The datasets provided to the participants for the scoring of the methods participating in T1 and T2 also included experimental trajectories of mRNA molecules in bacterial cells, telomeres in the cell nucleus, proteins in the cell membrane and cytoplasm, single atoms in an optical trap, and tracer particles in the cell cytoplasm and stirring liquid, from previously published works. For these trajectories, no objective ground truth is available besides the interpretation given in the literature. Therefore, it is not possible to assess their absolute errors and they were not included in the scoring. However, we found it interesting to carry out a comparative analysis of the predictions blindly provided by the 5 top-scoring challenge

participants in each task. Out of the whole dataset, we discuss the results for 4 representative experiments[20,38,47–49] for the inference of $\alpha$ (Fig. 5a–d) and the classification of the underlying model (Fig. 5e–h). The results obtained by all methods are shown in Supplementary Figs. 21–28.

The first dataset includes 2D trajectories of mRNA molecules inside live E. coli cells from the work by Golding and Cox[47] (Fig. 5a). Together with Ref. [50], these data provide one of the first evidence of subdiffusion in cellular systems. These experiments have generated a lively discussion about their underlying diffusion model (mainly between FBM and CTRW) and ergodicity[21,51–53]. All top-ranking methods provided distributions of exponents centered (median between 0.75 and 0.81) around the value estimated in the original publication ($\alpha = 0.77$) with variable width (st. dev. between 0.04 and 0.18) (Fig. 5e). However, the methods agreed in classifying the large majority

(between 74% and 100%) of trajectories as ATTM (Fig. 5i). This classification confirms the occurrence of ergodicity breaking, since both CTRW and ATTM are compatible with non-ergodic behavior and both have power-law waiting-time distribution. The preference toward ATTM might arise because of its varying diffusivity that better accounts for heterogeneity due to the biological environment or to variable noise.

The second dataset of experiments includes 2D trajectories of telomeres in the nucleus of mammalian cells[38,48] (Fig. 5b). It was previously shown that their TA-MSD features a FBM-like subdiffusive scaling for short and intermediate times with a mean exponent $\alpha \simeq 0.5$, approaching a linear behavior ($\alpha \simeq 1$) at longer timescales[48]. Also in this case, the classification methods largely agree and associate most of the trajectories to FBM (between 65% and 85%) (Fig. 5j). However, the determination of the exponent often produces a bimodal distribution with median values between 0.61 and 0.75 (Fig. 5f). Likely, the methods are not able to pick up the crossovers between diffusion regimes and rather assign an average exponent to each trajectory. The analysis of these experiments deserves the further methodological effort, since heterogeneous diffusion is emerging as a key feature of random motion in the biological environment[54].

The third dataset consists of 2D trajectories recorded for receptors diffusing in the plasma membrane of mammalian cells (Fig. 5c). In the original work[20], the TA-MSD was found to scale roughly linearly, whereas the EA-MSD showed subdiffusion with $\alpha \simeq 0.84$; this non-ergodicity was attributed to a temporal change of diffusivity and associated to ATTM. Once more, the classification methods largely confirmed previous results. A large percent of trajectories were attributed to the two models with time-dependent diffusion coefficients, namely the ATTM (between 57% and 71%) and the SBM (between 22% and 33%) (Fig. 5k). Moreover, inference methods consistently detected a large heterogeneity in $\alpha$, including both sub- and superdiffusion, with a slightly subdiffusive overall value, the median between 0.86 and 0.95 (Fig. 5g), in agreement to the original study[20].

To demonstrate the applicability of these methods beyond biological systems and at different Spatio-temporal scales, we included a dataset with 1D trajectories obtained for single atoms moving in a 1D periodic potential and interacting with a near-resonant light field that acts as a thermal bath[49] (Fig. 5d). These data were originally interpreted as evidence of CTRW with $\alpha = 1$[49]. Subsequently, the CTRW was deduced from microscopic parameters reproducing the trajectories without free parameters[55]. Because of the intrinsic complexity of this experiment, the trajectories were extremely short ($\approx 10$ data points), a regime that challenges the predictive power of any approach. Indeed, in this range of trajectory length all the methods showed rather large uncertainties on simulated data (Fig. 2f and Fig. 3f). However, since the microscopic mechanisms are well known, we aimed at using these experiments as a benchmark to check the predictive limits of the different approaches for very short trajectory length in a real scenario. The top regression methods for such short trajectories in 1D provided distributions spread over a wide range of $\alpha$, with medians between 0.8 and 0.91 (Fig. 5h). The results of model classification were also less conclusive with respect to the previous cases, likely a consequence of having short trajectories and of having $\alpha \simeq 1$, a regime where detectable differences among models are reduced (as shown in Fig. 3d). Predictions might also suffer from the lack of training data based on the microscopic model of Ref. [55], of which CTRW with $\alpha = 1$ is an approximation. Still, the CTRW was the most-likely model for 4 of the 5 top-scoring methods (between 28% and 48%, Fig. 5l), thanks to the capability of these methods to extract information from the microscopic dynamics of the generative models and not only from the long-term properties of the trajectory and its MSD.

The methods participating in T3 were not initially planned to be applied to the analysis of experimental data, due to the lack of trajectories featuring changes of diffusion models and/or anomalous diffusion exponent with the availability of previous analysis for comparison. However, when applied to some of the experimental trajectories described above, they did not evidence a significant occurrence of changepoints, as expected.

## Discussion
The results of the AnDi Challenge (T1) show that the choice of the analysis method strongly affects the accuracy in the determination of the anomalous diffusion exponent $\alpha$, in particular for more challenging conditions. Most of the methods outperform the conventional TA-MSD, even for long trajectories. For each dimension, we could identify a group of methods with comparable performance that greatly improve the precision of the anomalous diffusion exponent with respect to the baseline provided by the classical estimation of the MSD. These approaches were all based on machine learning, so we can infer that machine-learning-based methods can go beyond classical statistics, probably because they can extract from the trajectories of complex models some information that is not easily assessed by classical statistics. Despite a little degradation of performance, top-ranking methods perform best also for short and noisy trajectories, as shown by the correlation between metrics calculated over a subset of trajectories ($L < 200$, $SNR = 1$) with respect to the same metrics obtained over the whole dataset (Supplementary Fig. 29). This is a major improvement for trajectory analysis, since it enables collecting information from short and noisy tracks (e.g., those obtained by SPT PALM[56]) and from time segments of trajectories exhibiting heterogeneous behavior, without further averaging. However, the aspect that mostly boosts the overall performance is the ability to extract the anomalous diffusion exponent (an intrinsic ensemble property) for non-ergodic models from single trajectories (Fig. 2d). Top-performing methods are capable of determining model properties usually obtained from ensemble averages or feature distributions from patterns present in single trajectories. It is quite remarkable that this is possible even in the presence of noise that is known to hide non-ergodic behavior in some classical estimators[57] or with short trajectories that limit obtaining sufficient statistics for features such as the waiting-time distribution. This is a major limitation for approaches based on classical statistics (e.g., Bayesian inference) with models having several hidden variables that need to be systematically integrated. The availability of reliable methods to infer $\alpha$ will encourage researchers to further investigate the deviations from Brownian behavior that emerge in many experiments of interest, e.g., for biology and physics.

The AnDi challenge (T2) has led to the first concerted effort to develop methods able to classify individual trajectories among several mathematical models of diffusion. Machine-learning methods ranked top in the leader board and achieved an overall accuracy greater than 80% at detecting the ground-truth diffusion models. The comparison of $F_1$-score and AUC/ROC (Supplementary Fig. 17, Supplementary Fig. 18, Supplementary Fig. 19, and Supplementary Fig. 20) shows that most of the methods are quite confident at providing the correct classification. However, a limitation of all these classification approaches is that they can only choose among the diffusion models provided in the training. To robustly extend model classification to actual experiments, it can be useful to further widen the palette of models (e.g., by using ad hoc models), include a none-of-the-above class, and/or to include some metric of the confidence of the estimation (e.g., by using an entropy measure calculated on the predictions of an ensemble of machine-learning models).

Trajectory segmentation (T3 of the AnDi challenge) has been widely investigated when changes occur with respect to an estimator of the observable such as the mean or the variance[27]. Determining changes of anomalous diffusion is a rather novel problem, triggered by recent experimental findings[6,25]. We kept the challenge design rather simple, with trajectories of fixed length featuring exactly one changepoint. Even in this simple condition, the wide parameter space made the problem rather challenging, limiting the participation to T3 to only 4 teams. Yet, the submitted results showed an interesting asymmetry: The changepoint localization precision seems not only to depend on the relative length of the segments but also on the changepoint location (Fig. 4a), producing a lower RMSE for changepoints located at the beginning of the trajectory. Similarly, the methods show the best performance in estimating $\alpha$ and diffusion model for the first segment (Fig. 4b, c). We believe that this is at least partly a consequence of the inaccurate localization of the changepoint and the non-stationarity of some models. The inexact localization of the changepoint produces two spurious segments, altering the tail of the first segment and the initial point of the second by removing or adding spurious points. For non-stationary models, the initial point encloses information about the initiation of the physical process, thus improper segmentation impacts more severely the evaluation of the second segment[58].

From the blind analysis of various experimental datasets, we observed that the top methods, although based on different principles, lead to very similar results. This is encouraging as it points to an objective underlying reality of the anomalous diffusion phenomena and its mechanisms, which can be measured experimentally and has now been underpinned by the results of the AnDi challenge. Importantly, the results provided by the challenge methods were also in line with the conclusions of previous studies[20,38,47–49], further reinforcing their reliability. Interestingly, while the original works required a combination of several estimators, including ensemble averages, the challenge methods were able to provide compatible predictions in a one-shot analysis and with no prior knowledge about the experimental conditions. This is a particularly remarkable result, since the methods were not specifically trained to work with parameters used in experiments. In fact, experimental trajectories often show broad distributions of diffusion coefficients. In spite of a fixed localization error, this produces a non-uniform SNR with respect to our simulations. Also, experiments have different sampling rates with respect to the characteristic diffusion timescale. Accounting for the variability introduced by these effects during the training might improve the methods' prediction capability, further boosting their performance.

The number of experiments producing individual random trajectories is steadily increasing, accompanied by the production of ad hoc analysis tools. The AnDi challenge gave the opportunity to obtain a first assessment of some of these tools, oriented at detecting anomalous diffusion. In particular, we focused on methods quantifying deviation from pure Brownian behavior in terms of anomalous diffusion exponent and the underlying mathematical model. However, similar experiments are often analyzed following a more phenomenological approach, e.g., the classification of motion as diffusive, immobile, confined, or directed. Although the latter classification offers a more intuitive interpretation of random motion occurring in some systems, the models included in the challenge are strictly connected to these diffusion modalities. In fact, they allow a generalization of anomalous diffusion beyond the life sciences and include macroscopic natural and human processes, ranging from the foraging of animals to the spread of diseases, to trends in financial markets and climate records.

Building on these considerations, we believe it is necessary to establish clear and unified guidelines to identify and report anomalous diffusion, in particular from experiments, where the ground truth is not known. Possibilities in this sense might involve a list of key parameters to be quantified together with their respective confidence interval, e.g., based on the comparative use of multiple methods, involving both machine learning and classical statistics. The joint approach will allow to combine of advantages from both worlds: while machine learning methods are becoming more available and powerful, they often operate as a black box; estimators based on classical statistics can thus help to provide deep insight on anomalous diffusion phenomena.

The AnDi challenge gathered a large part of the community to trigger this discussion and collaborate on this unifying task. We hope this effort might be extended in the future to reach a larger consensus. To this aim, we have built an interactive tool (http://andi-challenge.org/interactive-tool/) where datasets and results of the challenge are stored; new methods can undergo an automated benchmarking according to the challenge rules and compare their scores with those of other participants. In fact, since the conclusions of the challenge, several participants have already improved their scores. Therefore, the challenge is permanently open and performance improvements will be continuously updated on demand.

## Methods

**Organization of the challenge.** We ran the Anomalous Diffusion (AnDi) challenge as a time-limited competition from March 1, 2020, to November 1, 2020. The competition was hosted on the Codalab platform (https://competitions.codalab.org/competitions/23601) and divided in three phases (Development, Validation, and Challenge). The competition has later been converted to an open challenge, continuously accepting new submissions. Datasets, methods, list of participants, and results of the AnDi Challenge are available at http://andi-challenge.org. Software for simulation and analysis is hosted on the competition GitHub repository https://github.com/AnDiChallenge.

**Challenge methods.** Among the participants, we could distinguish fifteen substantially different approaches (Table 1 and Supplementary Note 1). We classify the approaches based on three different criteria, as detailed in Table 1. First, we group methods based on the type of approach used, whether involving machine-learning or classical statistics. A large majority of methods are based on machine-learning architectures, such as recurrent neural networks (RNN), convolutional neural networks (CNN), gradient boosting machines, graph neural networks, extreme learning machines (ELM), or sequence learners. Other methods are based on statistical approaches, such as Bayesian inference, temporal scaling, and random interval spectral ensemble (RISE). A second grouping involves the type of input data used. Some methods employed feature engineering using classical statistics as an input, whereas other were simply fed raw trajectories. A further classification is based on whether methods required a specific training or model for different (ranges of) trajectory lengths (length-specific) or not. Several methods could be directly used or easily adapted to run multiple tasks.

**Structure of the datasets.** Simulated datasets were composed of synthetic trajectories generated according to five different mathematical models, both ergodic and non-ergodic: annealed transient time motion (ATTM, weakly non-ergodic), a motion with random changes of the diffusion coefficient in time[14], continuous-time random walk (CTRW, weakly non-ergodic), a motion undergoing local trapping with a wide distribution of waiting for times[11], fractional Brownian motion (FBM, ergodic), a motion with long-range correlated steps, often used to describe viscoelastic effects[12], Lévy walk (LW, ultra-weakly non-ergodic), a motion displaying irregular jumps with constant velocity, often associated with animal foraging strategies[13], and scaled Brownian motion (SBM, weakly non-ergodic), a motion whose diffusion coefficient features deterministic time-dependent changes[15]. We considered trajectories with anomalous diffusion exponents in the range $\alpha \in [0.05, 2]$. Exponents were restricted to $\alpha \geq 0.05$ because smaller exponents produce practically immobile trajectories. Note that CTRW and ATTM are strictly subdiffusive ($\alpha \leq 1$), LW is superdiffusive ($\alpha \geq 1$), FBM cannot have ballistic behavior ($\alpha < 2$), whereas SBM covers the whole exponent range.

Each dataset contained $10^4$ trajectories of variable length. All trajectories were first generated with a length $L = 1000$. For theoretical models providing trajectory sampling at irregular times (CTRW and LW), oversampling was used to obtain tracer coordinates at uniform times. The trajectories were then standardized to have a unitary standard deviation $\sigma_D$ of the distribution of displacements over unit time. To mimic experimental data, trajectories were corrupted with a finite localization precision. For this, a random number from a normal distribution $\mathcal{N}(0, \sigma_{\text{noise}})$ was added to each trajectory coordinate. Last, the displacements'

standard deviation was scaled by a random number sampled from a normal distribution $\mathcal{N}(0, 1)$ to include the effect of an effective diffusion coefficient (see Fig. 1a–c for exemplary trajectories in each dimension). Trajectories were thus cut to the desired length. For T1 and T2, trajectories were cut to lengths $L \in [10, 1000]$, whereas for T3 all trajectories had length $L = 200$. A different dataset was generated for each task to ensure the proper balance of the feature to be determined. Therefore, the dataset for T1 had a balanced distribution of anomalous exponents but not of diffusion models, whereas the dataset for T2 was balanced with respect to the diffusion models. For T3, trajectories were obtained by concatenating trajectories simulated for all models and exponents. Each trajectory had a random changepoint at a discrete index $t_{GT} \in [1, 199]$ corresponding to a change at least in one of the two features ($\alpha$ and diffusion model). An example of such kind of trajectories is presented in Fig. 1c.

Three levels of noise were used to corrupt trajectories, corresponding to $\sigma_{noise} = 0.1, 0.5, 1$. The SNR was calculated as SNR $= \sigma_D/\sigma_{noise}$, where $\sigma_D$ is the standard deviation of the distribution of displacements over unit time. Due to the previous standardization, the SNR levels were thus SNR $= 1, 2, 10$. Trajectories in 2D and 3D were allowed to have different noise levels along with different directions. The overall SNR was calculated as the average of SNRs calculated along with orthogonal directions.

We developed the `andi-datasets` Python package[59] to allow participants to generate their own dataset (e.g., for training). Examples of trajectories for various exponents and models are presented in Fig. 1c. Details about available functions can be found in the hosting repository https://github.com/AnDiChallenge/ANDI_datasets.

**Theoretical models**. In this section, we present a brief introduction to the concepts of anomalous diffusion and ergodicity breaking. We provide theoretical insights about the anomalous diffusion models considered in the AnDi challenge, as well as the description of the pseudocode used for simulations in 1D. Finally, we describe how to extend the algorithms to simulate the diffusion models in 2D and 3D, since for some models this is not simply obtained as the composition of motion along with independent directions. The Python implementation of all the algorithms described below is available at https://github.com/AnDiChallenge/ANDI_datasets[59].

*Anomalous diffusion and ergodicity breaking*. When analyzing trajectories, diffusion is typically quantified through the calculation of the mean squared displacement (MSD). The MSD grows linearly in time for Brownian walkers, MSD $\sim t$, while it shows a power-law scaling for anomalous diffusion, MSD $\sim t^\alpha$, where $\alpha$ is the anomalous diffusion exponent. In practice, the MSD can be calculated either by performing an ensemble average of the positions of a set of $N$ tracers,

$$\text{EA-MSD}(t) = \frac{1}{N}\sum_{i=1}^{N}[\mathbf{x}_i(t) - \mathbf{x}_i(0)]^2, \qquad (1)$$

or, for the trajectory of a single tracer, sampled at $L$ discrete times $t_i = i\Delta t$, as a time-average:

$$\text{TA-MSD}(\Delta = m\Delta t) = \frac{1}{L-m}\sum_{i=1}^{L-m}[\mathbf{x}(t_i + m\Delta t) - \mathbf{x}(t_i)]^2. \qquad (2)$$

In its most general definition, a process is considered ergodic if any single realization is able to explore all the possible configurations of the system. The impossibility of performing such an exploration is usually referred to as ergodicity breaking. For a (strong) non-ergodic process, the space of configurations is separated into mutually inaccessible domains, hence preventing its full exploration. If those domains are indeed accessible, but a single tracer is unable to visit them in a finite time, the process is instead defined as weakly non-ergodic[60]. In this case, a sufficiently large ensemble of tracers may indeed explore all possible configurations, hence producing a difference between ensemble and time averages.

In the context of anomalous diffusion, a system is said to show weak ergodicity breaking if the TA-MSD does not converge to EA-MSD in the infinite time limit[4]. Generally, while the EA-MSD still shows a power-law scaling, the TA-MSD scales linearly with the timelag[4]. Moreover, the value of the TA-MSD for different trajectories at a given time lag is a random variable, whose distribution can be analytically calculated for some diffusion models[61]. One can then define the time and ensemble-averaged TEA-MSD over a set of $N$ trajectories as

$$\text{TEA-MSD}(\Delta) = \frac{1}{N}\sum_{i=1}^{N}\text{TA-MSD}(\Delta)_i, \qquad (3)$$

where TA-MSD$(\Delta)_i$ is the TA-MSD for the $i$-th trajectory. The so-called ergodicity breaking parameter (EB)[51] can be calculated as

$$\text{EB} = \langle\zeta^2\rangle - 1, \qquad (4)$$

where $\zeta = \text{TA-MSD}(\Delta)/\text{TEA-MSD}(\Delta)$. The EB parameter, in the limit $\Delta/T \to 0$, is a widely used tool to quantify ergodicity breaking (here $T = L\Delta t$ represents the trajectory length). For ergodic diffusion, then EB $\to 0$, while any other value showcases a non-ergodic behavior. Processes like CTRW, ATTM, and SBM show weak ergodicity breaking[14,62,63], whereas Brownian motion and FBM are ergodic, though convergence of the EA-MSD to the TA-MSD may be slow for certain values of the anomalous exponent $\alpha$[64]. Indeed, as discussed in[24], the ergodicity of FBM

requires careful analysis as a function of $\alpha$, and often other statistical measures are necessary to study ergodicity breaking. To find a technique to study short trajectories, it is important to note that, for CTRW and ATTM, the TA-MSD shows a short-time linear behavior TA-MSD $\propto \Delta$ even for anomalous trajectories. This showcases one of the limitations of the fitting of the TA-MSD to determine the anomalous diffusion exponent. For the case of LW, a different kind of ergodicity breaking named ultraweak can been identified, where time and ensemble averages only differ by a constant factor[65,66].

*Continuous time random walk*. The continuous-time random walk (CTRW) defines a large family of random walks with arbitrary displacement density for which the *waiting time*, i.e., the time between subsequent steps, is a stochastic variable[11]. Here, we consider a specific case of CTRW for which waiting times are sampled from a power-law distribution $\psi(t) \sim t^{-\sigma}$ and displacements are sampled from a Gaussian distribution with variance $D$ and zero mean. In such case, the anomalous diffusion exponent is $\alpha = \sigma - 1$ (the EA-MSD = $\langle\mathbf{x}(t)^2\rangle \propto t^\alpha$). Since the waiting times are generated from a power-law distribution, for $\sigma = 2$ the EA-MSD features Brownian diffusion with logarithmic corrections[2]. For $\alpha = 1$ one should instead use a Poisson density, or a fixed waiting time (i.e., the limit of a one-sided Lévy stable density in the limit $\alpha = 1$).

The algorithm used to simulate CTRW trajectories is described in Algorithm 1. Notice that the variable $\tau$ stands for the total time at $i$-th iteration. Also notice that the output vector $\vec{x}$ corresponds to the position of the particle at the irregular times given by $\vec{t}$.

**Algorithm 1**. Generate CTRW trajectory

**Input:**
 length of the trajectory $T$
 anomalous exponent $\alpha$
 diffusion coefficient $D$
**Define:**
 $\vec{x} \to$ empty vector
 $\vec{t} \to$ empty vector
 $N(\mu, s) \to$ Gaussian random number generator with mean $\mu$ and standard deviation $s$
$i = 0; \tau = 0$
**While** $\tau < T$ **do**
 $t_i \leftarrow$ sample randomly from $\psi(t) \sim t^{-\sigma}$
 $x_i \leftarrow x_{i-1} + N(0, \sqrt{D})$
 $\tau \leftarrow \tau + t_i$
 $i \leftarrow i + 1$
**end while**
**Return:** $\vec{x}$, $\vec{t}$

*Fractional Brownian motion*. In fractional Brownian motion (FBM), $x(t)$ is a Gaussian process with stationary increments. This process is symmetric, $\langle x(t)\rangle = 0$, and importantly its EA-MSD scales as $\langle x(t)^2\rangle = 2K_H t^{2H}$. Here, $H$ is the Hurst exponent, which is related to the anomalous diffusion exponent as $H = \alpha/2$[12,67]. Also, the two-time correlation is $\langle x(t_1)x(t_2)\rangle = K_H(t_1^{2H} + t_2^{2H} - |t_1 - t_2|^{2H})$.

FBM can also be introduced as a process arising from a generalized Langevin equation where the noise is non-white (aka fractional Gaussian noise, fGn). The fGn has a standard normal distribution with zero mean and power-law correlations:

$$\begin{aligned}<\xi_{fGn}(t_1)\xi_{fGn}(t_2)> = {}& 2K_H H(2H-1)|t_1 - t_2|^{2H-2} \\ & + 4K_H H|t_1 - t_2|^{2H-1}\delta(t_1 - t_2).\end{aligned} \qquad (5)$$

The FBM features two regimes: one where the noise is positively correlated ($1/2 < H < 1$, i.e., $1 < \alpha < 2$, superdiffusive) and one where the noise is negatively correlated ($0 < H < 1/2$, i.e., $0 < \alpha < 1$, subdiffusive). For $H = 1/2$ ($\alpha = 1$) the noise is uncorrelated, hence the FBM converges to Brownian motion.

For a $d$-dimensional FBM, the corresponding position vector has zero mean, $\langle\mathbf{x}(t)\rangle = 0$, the EA-MSD is $\langle\mathbf{x}(t)^2\rangle = 2dK_H t^{2H}$, the autocorrelation is $\langle\mathbf{x}(t_1)\mathbf{x}(t_2)\rangle = dK_H(t_1^{2H} + t_2^{2H} - |t_1 - t_2|^{2H})$, and the fGN reads

$$\begin{aligned}<\xi_{fGn,i}(t_1)\xi_{fGn,j}(t_2)> = {}& 2K_H H(2H-1)|t_1 - t_2|^{2H-2}\delta_{ij} \\ & + 4K_H H|t_1 - t_2|^{2H-1}\delta(t_1 - t_2)\delta_{ij},\end{aligned} \qquad (6)$$

where $i, j$ in the subindex of the fGN denotes a different cartesian coordinate.

Various numerical approaches have been proposed to solve the FBM generalized Langevin equation exactly. Here, we use the Davies-Harte method[68] and the Hosking method[69] via the `FBM` Python package (https://pypi.org/project/fbm/). Details about the numerical implementations can be found in the associated references.

*Lévy walk*. The Lévy walk (LW) is a particular case of CTRW. The time between steps is irregular[13], but, in contrast to the CTRW considered here, the distribution of displacements for an LW is not Gaussian. We considered the case in which the flight times (i.e., the times between steps) are retrieved from the distribution

$\psi(t) \sim t^{-\sigma-1}$. In one dimension, the displacements are $\Delta x$ and the step length is $|\Delta x|$. The displacements are correlated with the flight times such that the probability to move a step $\Delta x$ at time $t$ and stop at the new position to wait for a new random event to happen is $\Psi(\Delta x, t) = \frac{1}{2}\delta(|\Delta x| - vt)\psi(t)$, where $v$ is the velocity. From here, one can show that the anomalous exponent is given by

$$\alpha = \begin{cases} 2 & \text{if } 0 < \sigma < 1 \\ 3 - \sigma & \text{if } 1 < \sigma < 2. \end{cases} \tag{7}$$

The details of the numerical implementation for the LW are given in Algorithm 2. Notice that we use a random number $r$, which can take values 0 or 1, to decide in which sense the step is performed. Also note that, as for the CTRWs, the output vectors $\vec{x}$, $\vec{t}$ represent irregularly sampled positions and times.

**Algorithm 2.** Generate LW trajectory
 **Input:**
 length of the trajectory $T$
 anomalous exponent $\alpha$
 **Define:**
 $\vec{x} \rightarrow$ empty vector
 $\vec{t} \rightarrow$ empty vector
 $v \rightarrow$ random number $\in (0, 10]$
 $i = 0$
 **While** $\tau < T$ **do**
 $t_i \leftarrow$ sample randomly from $\psi(t) \sim t^{-\sigma-1}$
 $x_i \leftarrow (-1)^r vt_i$, where random $r$ is 0 or 1 with equal probability.
 $\tau \leftarrow \tau + t_i$
 $i \leftarrow i + 1$
 **end while**
 **Return:** $\vec{x}$, $\vec{t}$

*Annealed transient time motion.* The annealed transient time motion (ATTM) implements the motion of a Brownian particle whose diffusion coefficient varies in time[14]. The tracer performs Brownian motion for a random time $t_1$ with a random diffusion coefficient $D_1$, then for $t_2$ with $D_2$, etc. The diffusion coefficients are sampled from a distribution such that $P(D) \sim D^{\sigma-1}$ with $\sigma > 0$ as $D \rightarrow 0$ and that decays rapidly for large $D$. If the random times $t$ are sampled from a distribution with expected value $E[t|D] = D^{-\gamma}$, with $\sigma < \gamma < \sigma + 1$, the anomalous diffusion exponent is $\alpha = \sigma/\gamma$ (corresponding to the subdiffusive *regime I* of the model described in Ref. [14]). Here, we consider that the distribution is a delta function, $P_t(t|D) \sim \delta(t - D^{-\gamma})$. Hence, the period of time $t_i$ in which the particle performs Brownian motion with a random diffusion coefficient $D_i$ is $t_i = D_i^{-\gamma}$, with $D_i$ extracted from the distribution described above. The numerical implementation of the ATTM model is given in Algorithm 3. Note that, in contrast to CTRW and LW, now the only output is $\vec{x}$ because the trajectory is already produced at regular time intervals of duration $\Delta t$.

**Algorithm 3.** Generate ATTM trajectory
 **Input:**
 length of the trajectory $T$
 anomalous exponent $\alpha$
 sampling time $\Delta t$
 **Define:**
 **while** $\sigma > \gamma$ and $\gamma > \sigma + 1$ **do**
 $\sigma \leftarrow$ uniform random number $\in (0, 3]$
 $\gamma = \sigma/\alpha$
 **end while**
 $BM(D, t, \Delta t) \rightarrow$ generates a Brownian motion trajectory of length $t$ with diffusion coefficient $D$, sampled at time intervals $\Delta t$
 $\vec{x} \rightarrow$ empty vector
 **while** $\tau < T$ **do**
 $D_i \leftarrow$ sample randomly from $P(D) D^{\sigma-1}$
 $t_i \leftarrow D_i^{-\gamma}$
 number of steps $N_i = \text{round}(t_i/\Delta t)$
 $x_i, ..., x_{i+N_i} \leftarrow BM(D_i, t_i, \Delta t)$
 $i \leftarrow i + N_i + 1$
 $\tau = \tau + N_i \Delta t$
 **end while**
 **Return:** $\vec{x}$

*Scaled Brownian motion.* The scaled Brownian motion (SBM) is a process described by the Langevin equation with a time-dependent diffusivity $K(t)$

$$\frac{dx(t)}{dt} = \sqrt{2K(t)}\xi(t), \tag{8}$$

where $\xi(t)$ is white Gaussian noise[15]. For the case in which $K(t)$ has a power-law dependence with respect to $t$ such that $K(t) = \alpha K_\alpha t^{\alpha-1}$, the EA-MSD follows $\langle x^2(t) \rangle_N \sim K_\alpha t^\alpha$ with $K_\alpha = \Gamma(1+\alpha)K_\alpha$. The numerical implementation of SBM is presented in Algorithm 4.

**Algorithm 4.** Generate SBM trajectory
 **Input:**
 length of the trajectory $T$
 anomalous exponent $\alpha$
 **Define:**
 $\text{erfcinv}(\vec{a}) \rightarrow$ Inverse complementary error function of $\vec{a}$
 $U(L) \rightarrow$ returns $L$ uniform random numbers $\in [0, 1]$
 **Calculate:**
 $\vec{\Delta x} \leftarrow (1^\alpha, 2^\alpha, ..., T^\alpha) - (0^\alpha, ..., (T-1)^\alpha)$
 $\vec{\Delta x} \leftarrow 2\sqrt{2}U(L)\vec{\Delta x}$,
 $\vec{x} \leftarrow \text{cumsum}(\vec{\Delta x})$.
 **Return:** $\vec{x}$

*Simulations in higher dimensions.* The algorithms presented above provide examples for the simulation of 1D trajectories. In order to maintain the properties of each anomalous diffusion model, extension to 2D and 3D was performed differently depending on the considered model. For ATTM, CTRW, FBM, and SBM in 2D, trajectories were obtained by the simple composition of (independent) motion performed over orthogonal axes. The same was done for FBM and SBM in 3D. For ATTM and CTRW (3D), and for LW (2D and 3D), waiting times and displacement lengths were sampled according to the recipe provided by each particular model in 1D. However, the displacement length was used to sets the radius of the circle (2D) or the sphere (3D) over which the tracer step ended up. The direction was randomly chosen to ensure the uniform sampling of the circle or the sphere, and coordinates along orthogonal axes were calculated accordingly.

**Metrics.** We calculated several metrics to quantify the performance of the submitted methods with respect to the ground truth in the various tasks. Although only the most representative metrics were used to build the competition leaderboard, others were used to gain further insight about the methods. We further built an interactive tool (http://andi-challenge.org/interactive-tool/) for comparing method performance (Supplementary Fig. 1). This application also provides a useful tool for developers to benchmark new methods.

*Challenge metrics.*

- Mean absolute error (MAE). Methods were required to provide an accurate prediction for the anomalous diffusion exponent $\alpha$ for a single trajectory (T1) or for a part of a trajectory after segmentation (T3). Method performance was thus quantified by the MAE between the predicted value and the ground truth:

$$\text{MAE} = \frac{1}{N}\sum_{i=1}^{N}|\alpha_{i,\text{p}} - \alpha_{i,\text{GT}}|, \tag{9}$$

where $N$ is the number of trajectories in the dataset, and $\alpha_{i,\text{p}}$ and $\alpha_{i,\text{GT}}$ represent the predicted and ground-truth values of the anomalous exponent of the $i$-th trajectory, respectively.

- $F_1$-score. For T2 and T3, the methods have to provide a score of the probability for a trajectory (or a segment) to be assigned to one of the five diffusion models. Predictions for which the highest probability value corresponded to the ground-truth model were identified as true positives. As a summary statistics for model classification, we used the $F_1$-score. For multi-class classification problems, scoring metrics such as precision, recall, and $F_1$-score can be computed as a macro-average (which evaluates the metric independently for each class and then take the average, giving all classes the same weight), or as a micro-average (which aggregates the contributions of all classes to compute the average metric). Micro-averaging is generally preferable when class imbalance is present. Although the challenge was based on a balanced dataset with each class equally represented, we used a micro-averaged $F_1$-score in order not to provide any hint to participants about the content of the dataset. The micro-averaged $F_1$-score was calculated as

$$F_1 = \frac{2\,\text{TP}}{2\text{TP} + \text{FP} + \text{FN}}, \tag{10}$$

where TP, FP, and FN represent true positives, false positives, and false negatives calculated over the whole dataset, respectively.

- Root mean square error (RMSE). The trajectory segmentation problem in T3 requires the location of the point where a trajectory undergoes a change in anomalous diffusion. The most important consideration for a changepoint method is how accurately it localizes the changepoint itself. The quantification of this accuracy was performed through the RMSE between the predicted and ground truth position:

$$\text{RMSE} = \sqrt{\frac{1}{N}\sum_{i=1}^{N}\left(t_{i,\text{p}} - t_{i,\text{GT}}\right)^2}, \tag{11}$$

where $t_{i,\text{p}}$ and $t_{i,\text{GT}}$ represent the predicted and ground-truth values of the changepoint position, respectively. Unlike for T1, where we used the MAE,

in this case, we opted for the RMSE. This quadratic metric gives a higher weight to large errors, thus penalizing methods that provide predictions very far from the ground truth.

- Mean reciprocal rank (MRR). For ranking purposes of T3, the precision in determining the changepoint position, the anomalous diffusion exponent $\alpha$, and the diffusion model were summarized into a single statistics for the overall method evaluation, given by the MRR:

$$\text{MRR} = \frac{1}{3}\left(\frac{1}{\text{rank}_{\text{MAE}}} + \frac{1}{\text{rank}_{F_1}} + \frac{1}{\text{rank}_{\text{RMSE}}}\right), \tag{12}$$

where $\text{rank}_{\text{MAE}}$, $\text{rank}_{F_1}$, and $\text{rank}_{\text{RMSE}}$ correspond to the position in an ordered list based on the value of the corresponding metrics. For this task, MAE and $F_1$-score were calculated by treating each segment (before and after the predicted changepoint) as an individual trajectory and averaging the metrics obtained over the two segments.

*Additional metrics.* Further statistics were used for the comparative analysis of the performance of the methods.

- Anomalous exponent bias. For the determination of the anomalous diffusion exponent in T1 and T3, besides the accuracy, we further assessed whether the predicted value systematically differed from the ground truth. For this reason, we calculated the distribution of the difference between predicted and ground truth exponent (Supplementary Fig. 14, Supplementary Fig. 15, and Supplementary Fig. 16), and estimated the bias $\theta$ as its expectation value:

$$\theta = \frac{1}{N}\sum_{i=1}^{N}(\alpha_{i,\text{p}} - \alpha_{i,\text{GT}}). \tag{13}$$

As shown in Fig. 2, the estimation of the anomalous diffusion exponent from the fit of the TA-MSD shows a negative bias (i.e., the predicted exponent $\alpha_\text{p}$ is systematically smaller than the ground truth exponent $\alpha_\text{GT}$). Such effect is particularly close to $\alpha_\text{GT} = 1$ and is associated to the presence of localization error[18]. However, as shown in Supplementary Fig. 14, Supplementary Fig. 15, and Supplementary Fig. 16, the top-performing methods show little or no bias in their predictions.

- Receiver operating characteristic (ROC) curve and area under the curve (AUC). The calculation of the $F_1$-score assumes that a method outputs a discrete classifier (i.e., a unique choice for the diffusion model). However, many methods output continuous numbers associated to the probability of the input to belong to each class. Thus, these values assigned to each model contain more information about the performance of the classifier. This information can be summarized by the ROC curve and the corresponding AUC. The ROC curve reports the true positive rate (or sensitivity) versus the false negative (one minus the specificity) for different levels of probability thresholds: if an input has a certain class probability above the threshold, it is considered to belong to such class. The AUC is given by the integral of the ROC curve and is equal to the probability that a classifier will rank a randomly chosen positive instance higher than a randomly chosen negative one. It thus provides a useful tool to compare the sensitivity and specificity of a given classifier. In particular, being based on probability instead of class labels, ROC/AUC reports how "doubtful" a method is about its choice of the model. ROC curves for each class versus the others are shown in Supplementary Fig. 17, Supplementary Fig. 18, and Supplementary Fig. 19 for all teams. Micro- (i.e., considering each class as a binary prediction) and macro-averaged (i.e., considering an equal weight for the classification of each label) ROC curves are also reported. The ROC/AUC analysis confirms that ATTM is the most problematic model to classify, whereas the best results are obtained for CTRW and LW. The scatter plot of values of $F_1$-score vs. micro-averaged AUC shows a rather good correlation (Supplementary Fig. 20), with the exception of a few models (teams L, D, and N) that perform considerably better in terms of $F_1$-score.

- Recall, false-positive rate, Jaccard similarity coefficient, and $\text{RMSE}_\text{TP}$. For the assessment of the changepoint localization error in T3, we followed two different evaluation approaches. For the challenge evaluation, we simply quantified the RMSE. Trajectories showing no changepoint were considered as having a dummy changepoint either at index 1 or 199. However, to get a better understanding of methods' performance, we also considered an alternative analysis. For this, trajectories with ground truth and predicted changepoints within a distance $\epsilon = 20$ from the start/endpoints were considered as not having a changepoint. We thus considered four cases:

  - predicted and ground-truth positions located at $\epsilon < t < L - \epsilon$, counted as true positive (TP);
  - predicted and ground-truth positions located at $t \leq \epsilon$ or $t \geq L - \epsilon$, counted as true negatives (TN);
  - the predicted position located at $\epsilon < t < L - \epsilon$ but the ground-truth located at either $t \leq \epsilon$ or $t \geq L - \epsilon$, counted as false positive (FP);

- the predicted position located at either $t \leq \epsilon$ or $t \geq L - \epsilon$, but the ground-truth located at $\epsilon < t < L - \epsilon$, counted as false negative (FN).

Based on this classification, we evaluated the recall (also known as sensitivity):

$$\text{recall} = \frac{\text{TP}}{\text{TP} + \text{FN}}; \tag{14}$$

the false positive rate:

$$\text{FPR} = \frac{\text{FP}}{\text{FP} + \text{TN}}; \tag{15}$$

and the Jaccard similarity coefficient (JSC) for binary classification:

$$\text{JSC} = \frac{\text{TP}}{\text{TP} + \text{FP} + \text{FN}}. \tag{16}$$

We also calculated the $\text{RMSE}_\text{TP}$, corresponding to the RMSE obtained only for prediction/ground-truth pairs classified as true positives.

### Alternative and baseline estimators

*Inference of the anomalous diffusion exponent.* Several classical statistical methods have been employed to characterize anomalous diffusion from single trajectories and quantify the anomalous diffusion exponent. Many of them rely on the analysis of the EA-MSD or TA-MSD presented in Eqs. (1) and (2).

We developed a simple tool to perform the estimation of the anomalous exponent to establish a performance baseline for T1 of the challenge. The code calculates the TA-MSD and performs a linear fit of its logarithm with respect to the logarithm of the timelag for the first $k$ data points, where $k$ is the maximum between 10 and 10% of the trajectory length. The anomalous diffusion exponent is thus derived as the slope of the straight line. This criterion has been shown to provide reliable results for the fitting of TA-MSD for Brownian diffusion[70]. Although the choice of a different timescale or the use of an independently calculated localization precision can produce better results[16], we intentionally limited the code to a simple fitting algorithm with a straightforward criterion for the choice of the number of data points to fit. As shown in Fig. 2d, for ergodic models (FBM), such a simple fit produces results comparable with the best methods. In addition, as it can be observed using the interactive tool (http://andi-challenge.org/interactive-tool/), the estimation of $\alpha$ through the fit of the TA-MSD even outscores the other methods for ergodic models (FBM) at the highest SNR (e.g., SNR = 10). The code is available at https://github.com/AnDiChallenge/ANDI_datasets[59].

For the sake of completeness, we would like to mention other statistical approaches not considered in the challenge that can be used to tackle T1. Besides the MSD, another popular methodology for the quantification of the anomalous diffusion exponent is the moment scaling spectrum MSS[71,72]. MSS considers several high-order moments of the displacement distribution to obtain their scaling exponents and uses them to calculate the slope of the exponent curve versus the moment order, which is found proportional to $\alpha$.

The anomalous diffusion exponent is strictly linked to specific characteristics of the diffusion model, thus it can also be obtained by means of their quantification[4]. However, this approach requires the knowledge (or an educated guess) of the diffusion model. If, in addition, distributions of the associated quantities can be obtained, then anomalous diffusion exponent can be estimated through their fitting. For instance, for CTRW, the anomalous diffusion exponent can be extracted by fitting the waiting time distribution $\psi(t)$[19]; for the ATTM, by fitting the distribution of diffusion coefficients or transit times[20]; or, for a Lévy walk from the flight time or step length distribution[73].

*Classification of the underlying diffusion model.* Even though the problem of associating a trajectory to an underlying diffusion model has been long investigated, there is still no clear general consensus on how to unambiguously determine the underlying physical mechanism from a trajectory. To the best of our knowledge, model classification is generally performed using a combination of multiple estimators and further corroborated by a comparison with the corresponding analysis of simulated data. Several statistical parameters have been proposed in this sense. Algorithms based on multiple estimators can allow distinguishing between pairs of models[21–23]. Some of the proposed approaches are based on trajectory statistical features to determine ergodicity[21,51] and Gaussianity[74], and thus restrict the number of possible models. Lastly, the velocity autocorrelation function[75] and the power spectral density[38] have been shown to have model-dependent fingerprints for some diffusion models. However, none of these methods can be directly used to classify the trajectories as required for T2. First attempts to provide a direct and generalized classification have been proposed only recently[36,41,43] and the developing teams have participated in the challenge. Therefore, we decided not to provide any baseline estimation for this task.

*Trajectory segmentation.* Although a few methods have been recently developed for the detection of trajectory changepoints with respect to a switch in $\alpha$[25,32,33] and diffusion model[34], there is no consensus on a well-established method that can be used as a baseline for T3. Limited to the changepoint detection part, we thus

decided to compare methods' performance with the results of a random prediction, as shown in dashed lines in Fig. 4a and Supplementary Fig. 5. For this, we simply calculate the RMSE for selecting a random point on a trajectory having a changepoint at $t_{GT}$. The error associated with such a random prediction is not uniform, since it depends on the changepoint position $t_{GT}$ along the trajectory, as well as on the trajectory length $L$. The random predictor $RMSE_{random}$ can thus be calculated as the RMSE for a trajectory with a changepoint at position $t_{GT}$ and random predictions $t$ of the changepoint drawn from a uniform distribution in the range $[0, L]$

$$\text{RMSE}_{\text{random}}(t_{\text{GT}}) = \sqrt{\frac{1}{L}\int_0^L \left(t - t_{\text{GT}}\right)^2} = \sqrt{\frac{t_{\text{GT}}^3 + \left(L - t_{\text{GT}}\right)^3}{3L}}, \quad (17)$$

where $L$ is the trajectory length.

**Reporting summary**. Further information on research design is available in the Nature Research Reporting Summary linked to this article.

## Data availability
The simulated data used in this study are available for download at the competition website http://andi-challenge.org/challenge2020/. Ground-truth datasets used in the first phase of the competition for training are also available.

## Code availability
All software used for the Challenge is available at https://github.com/AnDiChallenge. The code of the andi-datasets package[59] used to generate the competition datasets is available at https://github.com/AnDiChallenge/ANDI_datasets.

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

## Acknowledgements

The authors would like to thank: Paula Kowalek for the graphical illustrations; Matthias Weiss and Maria Garcia-Parajo for sharing experimental data; Daniel Adam for help with compiling the data of single-atom trajectories. G.M.-G., B.R., and M.L. acknowledge support from ERC AdG NOQIA, Agencia Estatal de Investigación "Severo Ochoa" Center of Excellence CEX2019-000910-S, Plan Nacional FIDEUA PID2019-106901GB-I00/10.13039/501100011033, FPI), Fundació Privada Cellex, Fundació Mir-Puig, and from Generalitat de Catalunya (AGAUR Grant No. 2017 SGR 1341, CERCA program, QuantumCAT U16-011424, co-funded by ERDF Operational Program of Catalonia 2014-2020), MINECO-EU QUANTERA MAQS (funded by State Research Agency (AEI) PCI2019-111828-2/10.13039/501100011033), EU Horizon 2020 FET-OPEN OPTOLogic (Grant No 899794), and the National Science Centre, Poland-Symfonia Grant No. 2016/20/W/ST4/00314. Giov.V. and A.A. acknowledge funding from ERC StG ComplexSwimmers (Grant No. 677511) and from the Knut and Alice Wallenberg Foundation. M.A.G.-M. acknowledges funding from the Spanish Ministry of Education and Vocational Training (MEFP) through the Beatriz Galindo program 2018 (BEAGAL18/00203). R.M. acknowledges DFG grant ME 1535/12-1. Gior.V. and A.G. acknowledge sponsorship for this work by the U.S. Office of Naval Research Global (Award no. N62909-18-1-2170). Z.H. acknowledges funding from the Fundamental Research Funds for the Central Universities. J.-H.J. acknowledges NRF grants 2020R1A2C4002490 and 2017K1A1A2013241. T.B. acknowledges support by the Francis Crick Institute, which receives its core funding from Cancer Research UK (FC001086), the UK Medical Research Council (FC001086), and the Wellcome Trust (FC001086), and thanks Nate Goehring for supervision and acquisition of funding. This research was funded in whole, or in part, by the Wellcome Trust (FC001086). For the purpose of Open Access, the author has applied a CC BY public copyright license to any Author Accepted Manuscript version arising from this submission. J.A.C. acknowledges support from the ALBATROSS project (National Plan for Scientific and Technical Research and Innovation 2017-2020, No. PID2019-104978RB-I00). P.K, H.L.-O. and J.S. were funded by the Polish National Science Centre (NCN-DFG Beethoven Grant No. 2016/23/G/ST1/04083) and acknowledge the support by the Wroclaw Centre for Networking and Super-computing (calculations were performed using their BEM computing cluster). S.T. acknowledges the Deutscher Akademischer Austauschdienst for PhD Scholarship (DAAD Program ID 57214224) and support in the form of a Sacker postdoctoral fellowship and funding from the Pikovski-Valazzi matching scholarship (Tel Aviv University). H.K. and I.S. acknowledge funding from the Dutch Research Council (NWO) through the GENOME-TRACK project of the Building Blocks of Life research program (Project No. 737.016.014). C.M. acknowledges funding from FEDER/Ministerio de Ciencia, Innovación y Universidades – Agencia Estatal de Investigación through the "Ramón y Cajal" program 2015 (Grant No. RYC-2015-17896), and the "Programa Estatal de I+D+i Orientada a los Retos de la Sociedad" (Grant No. BFU2017-85963-R); from the Generalitat de Catalunya (AGAUR Grant No. 2017SGR940). C.M. also acknowledges the support of NVIDIA

Corporation with the donation of the Titan Xp GPU and funding from the PO FEDER of Catalonia 2014-2020 (project PECT Osona Transformació Social, Ref. 001-P-000382).

## Author contributions

C.M. conceived the study. C.M., G.M.-G., Giov.V., M.A.G.-M, M.L., and R.M. organized the challenge and the corresponding workshop. G.M-G. designed and implemented the software for data generation and comparison of results. G.M.-G. generated the data and ground truth used in all challenge phases. G.M.-G. and C.M. verified the files submitted by the participants and performed the scoring of all methods. G.M.-G., C.M., Giov.V., and M.A.G.-M. analyzed the results. The methods were designed, implemented, run, and described by the participating teams: team A: B.R., G.M.-G.; team B: S.T., M.A.L., J.-H.J., S.P., Y.K.; team C: J.-B.M., H.V.; team D: T.S., C.B.H., J.-H.J.; team E: A.A., S.B.; team F: H.K., I.S.; team G: Z.H.; team H: N.F., J.A.C., O.G.O.; team I: C.M.; team J: T.B.; team K: E.A., P.G.M.; team L: Gior.V., A.G.; team M: O.G.O., J.A.C.; and teams N,O: H.L.-O., P.K., J.S. D.K., C.M., and A.W. provided experimental datasets. The article was written by C.M., G.M.-G., Giov.V., and M.A.G.-M. with input from all authors.

## Funding

## Competing interests

The authors declare no competing interests.
