## [Peer Review File · Nature Communications]

Objective comparison of methods to decode anomalous diffusionREVIEWER COMMENTS

Reviewer #1 (Remarks to the Author):

This manuscript describes the results of a competition to compare methods for classifying and characterizing various forms of anomalous diffusion. It follows in the footsteps of previous community-wide competitions, such as the particle tracking challenge (Nat. Methods 2014) and the cell tracking challenge (Nat. Methods 2017). These competitions are very helpful to the community, as they give the reader a good overview of what methods are available and how the methods perform, as judged by criteria applied identically to all methods. Importantly, each method is applied by its own developers/protagonists, giving it the best chance to succeed, thus eliminating concerns about potential suboptimal use due to lack of familiarity or incomplete understanding.

This challenge had good participation, with 15 teams applying different methods, some using machine learning, others using classical statistics, and some using a combination of the two. The challenge assessed the various methods' performance in three tasks: identifying the anomalous diffusion exponent α , determining the underlying diffusion model (choice of 5 different models), and detecting switches in diffusion properties within a single trajectory. These tasks were performed in 1D, 2D and 3D. Each participating team submitted results for whichever tasks and dimensions of interest/relevance to them.

The manuscript is overall well written, and I appreciated its combination of results presentation and detailed discussion of the different anomalous diffusion models, almost as a review/tutorial.

From my reading, the challenge resulted in quite a clear conclusion of which algorithms perform best, namely those based on machine-learning. This brings me to a critique of the abstract, where the conclusion is more open/vague. It is true, as the abstract says, that no single method performed best across all tasks. However, a group of methods did, and those were all based on machine learning. I believe this is a conclusion worth mentioning upfront in the abstract, so that a reader actually learns something from it. If other methods based on machine learning did not perform well, it is also worth discussing the difference between them and the more successful machine learning based methods. The purpose of these competitions is to guide both users and developers in terms of where to put their effort and energy, but this guidance can only happen if conclusions are stated clearly.

Task 1 (identifying α) and Task 2 (determining diffusion model) are well defined and the algorithms' performance is well characterized. Task 3 (trajectory segmentation in the case of switches) however is lacking an important aspect, namely the question of whether a changepoint is detected in the first place. The task as currently stated is concerned with the accuracy of determining the switching point, but it is not clear what happens if no switch point is detected at all. These are two separate aspects of this task that should be analyzed independently. First, whether a switch point is detected at all (in fact, one can ask about sensitivity (whether the real switch point is detected) and specificity (whether many false positive switch points are detected)). Second, for trajectories where the switch point is detected, one can then ask the question that is currently addressed, namely the accuracy of the switch point location. It is possible that with their current analysis these two questions are convoluted into one performance measure, but this may produce misleading results.

The analysis of recall vs. RMSE_TP in Fig. 4d needs better explanation.

Also regarding Task 3, I find the results in Fig. 4e and f rather surprising. For both, I would have expected the accuracy of switch point detection to be worst along the diagonal, because the trajectory would not really be switching its diffusion model/degree of anomalous diffusion. In addition, also for both, I would have expected some symmetry in these matrices in the off-diagonal. For example, in Fig. 4e, how come switching from $\alpha > 1.5$ to $\alpha < 0.5$ has an RMSE of 23.9, while switching from $\alpha < 0.5$ to $\alpha > 1.5$ (just the opposite of the previous switch) has a much worse RMSE of 36.7? As another example, how can switching from $\alpha < 0.5$ to $\alpha > 1.5$ have a worse RMSE than switching from $\alpha > 1.5$ to $\alpha > 1.5$ (36.7 vs. 29.3)? The

authors should investigate these results further, and discuss them in greater depth. There is some mention of asymmetry in the discussion (p15, lines 427-430), but that is too brief and does not even refer to Fig. 4e and f.

In terms of performance measures, why do the authors use MAE to compare predicted and ground truth alpha, but then use RMSE to compare the predicted and ground truth changepoint position? Both are comparing individual values to each other, and it is not clear why this difference in treatment, which adds yet another acronym for the reader to remember while reading the manuscript.

Comments on figures:

As a major critique of manuscript organization/data presentation, I believe that most applications of anomalous diffusion analysis are for 2D trajectories. For example, among the experimental data presented in the paper, 3 out of the 4 datasets are in 2D. Thus, I think it will be most useful to present the 2D results in the main figures, instead of the 1D results currently presented. This applies to Fig. 2d-g, Fig. 3d-g and Fig. 4 (all panels I think).

In Fig. 2a-c and Fig. 3a-c, it is not clear what the different color bars represent. The legend in Fig. 2 says "colors represent mean absolute error contributions for each underlying diffusion coefficient." But then what does it mean to stack them up? Should they not be averaged to get overall MAE? For example, what do the yellow bars in Fig. 2a mean? Does this mean the MAE for SBM is ~0.3 for say Team K, or is this just an artifact of stacking the bars, and their real MAE is actually < 0.1 ?

As far as I understand, the confusion matrix in Fig. 3e should be identical to the confusion matrix in Suppl. Fig. S11 for Team E (first confusion matrix in Suppl. Fig. S11). But they are different. Why? Overall, some of the confusion matrices in Figs. S11-S13 add up to 100 per row and per column, and some do not. Shouldn't all of them be adding up to 100 per row and per column?

Other comments:

p. 7: What exactly does micro-average refer to in the F1-score?

p.7: The authors should refer to Eq. 12 in Materials and Methods section when mentioning "mean reciprocal rank."

p. 8 "Participating teams and methods": When mentioning the grouping of approaches, it would help to explicitly list the different teams belonging to each approach.

p. 10, line 261: Clarify what is meant by "smoothly" in the phrase "performance varied more smoothly."

p. 12: Please mention dimensionality of experimental data trajectories. Yes this is mentioned in Supplementary Note 2, but that is far away in the manuscript and I don't think there is any reference to it from the main text.

p. 14, lines 393-394: The authors state that providing precise and unbiased prediction for short and noisy trajectories is important for a method's performance. But I am not sure how the authors conclude this from their data and analyses. From the figures, all methods are worse for shorter trajectories and better for longer trajectories. And all methods are worse for lower SNR and better for higher SNR. While this statement is kind of self-evident, I am not sure how it stems from the data and analyses of the manuscript, as implied by that statement.

p. 34, line 755: Reference to a non-existent Fig. 1d.

p. 35, line 788: The authors state that "the EA-MSD does not converge to TA-MSD in the infinite time limit." Should the statement not be the other way around, namely "the TA-MSD does not converge to EA-MSD in the infinite time limit." As the authors themselves state on p. 4,

“Traditionally, the MSD is defined as the ensemble average over a group of tracers.”

p. 35, line 789: This statement needs a reference.

p. 35, line 790: What is the “amplitude” of the TA-MSD?

p. 44, line 961: Please define “Matching changepoints.” Is there a tolerance, or is the match exact?

p. 45, line 980: The authors state “As shown in Fig. 2, such a simple fit produces results comparable with the best methods for ergodic models with high SNR and long trajectories.” Where is this shown in Fig. 2?

p. 45, lines 983 – 995: This text seems to be there only for completeness. This should be stated explicitly.

p. 46, Eq. 14: This equation needs either derivation or a reference.

Supplementary Note 1: The descriptions of the different algorithms are full of acronyms. Please define/spell out all new terms. Also, some methods state a reference, while others do not. This might be because the methods without a reference were developed for the purposes of the challenge. In this case, they should still state a reference, and it can say something like “this work.” This eliminates any ambiguity.

Reviewer #2 (Remarks to the Author):

Gorka Munoz-Gil et al., present a summary of how effective is the performance of different data analysis methods applied to stochastic trajectories, in particular those involving anomalous diffusion. The different analysis methods are from typical statistics (Bayesian inference) to machine learning. A competition called AnDi, was open to the community, and based on this one can gain new insights on the different methods, for example which performs better.

Trajectory data gained these days, from single molecule tracking to other fields like finance is clearly enormous. In many cases such paths are modelled with anomalous diffusion deviating from traditional Brownian motion. Usually, it is easy to tell from experimental data that Einstein's theory of Brownian motion fails. However, predicting the type of non-Brownian dynamics and even determining the exponent α , characterizing the mean square displacement, can be hard. Hence, I believe the authors have served a wide range of communities, by making this type of analysis through competition. I am not familiar with a similar scientific battle and hence the whole approach is timely and refreshing.

In short, it is my opinion that the authors promoted the field of data-analysis relevant to many fields.

The authors define three tasks:

Task 1 determine the anomalous diffusion exponent α .

Task 2 determine the underlying diffusion model.

Task 3, does the diffusion process change within the observation window (trajectory segmentation).

They then check which method performs better (see more details on the methods in the manuscript). I believe the definition of tasks are useful and reasonable. For example the value of α determines how fast certain molecules diffuse in the cell, and this is clearly important for bio-chemical reactions and in general for the ticking of the cell.

The authors show clearly how effective the different methods are for succeeding in the three different tasks, involving noise corrupted simulation data with predefined anomalous exponents and specific diffusion /transport models.

Particularly, the authors show how machine learning methods are better for determining the anomalous exponent in the data set from simulations (Task 1), even for short time series. For determining which diffusion model the simulation data is described by (Task 2) for a broad range of values of alpha. And Finally, for discriminating between the alpha exponent and diffusive models in time series cut randomly by half. In all of them Machine learning methods exhibit a better performance.

-- INSERT --

Plus the authors show the performance of the best competitors for solving Task 1 and Task 2 on experimental data sets for four different experiments. (diffusion of mRNA inside the cell, telomeres inside the nucleus, plasma receptors on the membrane and atoms in optical lattices). The majority of them were consistent with the values and models reported in the literature, just the second and last case shows discrepancies in the value of the anomalous exponent. More on this see below.

Good features

The paper is interesting and clearly exhibits how machine learning methods give better results if compared with classical statistical methods when time series with anomalous diffusion are studied. The comparison between different methods is insightful.

*** Some questions/remarks ***

The authors don't explain why Task 3 was not incorporated in the study of experimental data sets. Having a comparison between machine learning segmentation and for example local convex hull method could be useful.

Did we learn something new on the systems involved? for me a surprising result was the analysis of the motion of atoms in one dimensional optical potentials.

Based on theoretical arguments it was suggested previously that the process belongs to the CTRW class

of the normal type. The microscopical model in mind was an underdamped particle in a periodic structure, a perfectly normal model with $\alpha=1$. Now the authors show, based on available experimental data, that the system exhibits sub-diffusion (though errors are large, see fig. 5h). One could argue that this is a nice demonstration of the power of the methods involved. However, at a closer look we may suspect that the conclusion is simply misleading.

To see this recall that the training session did not include the stochastic model which was suggested in the literature.

See letter titled CTRW for a particle in a periodic potential PRL, 123, 070602 (2019) in particular please pay attention to the under-damped Langevin model (Eq. 2 there).

Note that the authors (Dechant et al.) in mentioned PRL, find excellent agreement between theory and experiment without any fitting parameter, which is most likely not a coincidence. Hence, I think authors of the present paper are making a daring claim.

To solve this mystery I suggest the authors meet the following challenge.

Please train the machine with the model in the mentioned PRL, and see what is the verdict with respect to the experimental data. I would not be surprised if the Langevin model, in the presence of periodic potential, works fine.

If so we may reach the conclusion that depending on the input given for the purpose of machine learning, we get vastly different pictures of data (however for now this is merely my speculation). Any way, it is hard for me to accept that a model without fitting parameters is simply wrong, and unfortunately I am in a state of confusion. As mentioned, I cannot say if this is a limitation of the previous analysis or the present one, and the authors can clarify this point by adding the requested model to the tool box.

In general the AnDi was built for models that can exhibit either anomalous or normal diffusion (alpha is a control parameter). What happens when you add

models which are perfectly normal, with no knob on alpha?

This is not just a nasty question, real data that is perfectly normal might be wrongly interpreted as anomalous.

In other words, how important is the input of five anomalous diffusion models (ATTM, CTRW, FBM, LW, and SBM)??

why these models? with respect to simulation competition this is fine, as one needs to start with certain models. But when analysing

real data, can we be sure that this choice of five models does not create a bias? for example over damped Langevin models in the presence of local fields is a reasonable model to start with. Why was it excluded?

To conclude, I find the paper very interesting and promising. Some clarification on the applications to real data is needed, while the competition shows clear insights (machine learning wins) with respect to simulated data.

Minor issues:

on line 113 authors write that the use of TA MSD is not valid, for the determination the exponent alpha.

While the intention of the authors is clear this sentence is too strong.

Simply said we have two ways to determine alpha, either with the time or with the ensemble average, though neither can be considered superior with respect

to the other. In the bulk of the paper the aim is to obtain the exponent of

the EA MSD, which is fine, but seems to me arbitrary, in fact in the realm of single molecule tracking, I think the opposite is natural.

line 185 explanation on the meaning of micro-averages is needed.

Trajectory segmentation, implies a sudden jump in the underlying control parameters, this is a special case of the more general time dependent change of the process (the latter category is of course endlessly large). I am not sure if any of the methods could deal with such a case.

Power spectrum analysis, convex hull methods, p-variation approach, were not used and not mentioned (unless I missed this). Is there a deep reason for this?

or is this simply because these methods were not represented in the competition?

Reviewer #3 (Remarks to the Author):

This is a very interesting paper describing the results of an interesting challenge developed to test different approaches to analyzing anomalous diffusion. Overall, I think this is a strong contribution of significant merit and warrants acceptance into Nature Methods [ED: all comments also relevant for Nature Communications]. I do, however, have a few comments/concerns/questions. They are organized below by section.

INTRODUCTION

- I am very happy to see the list of issues with using the MSD. Despite this, the paper elevates the estimation of the anomalous coefficient in terms of that seen on the MSD to be on par with determining the underlying model (and, presumably, its parameters). Can the authors provide additional justification for the continued use of the anomalous coefficient and the MSD? Does it provide information that is useful to the practitioner or is it simply the culture to look at data this way?

RESULTS

- line 205 states that algorithms were grouped into three classes (ML over raw, ML over stat, classical state) but the table does not present them this way, Rather, the table defines a column

for class (two choices), input (two choices), and L-specific (yes/no). The description should be harmonized to the table.

METHODS

- line 757 SNR was defined as $\sigma_D/\sigma_{\text{noise}}$ but I cannot find a definition of σ_D . It's probably somewhere but I'm not seeing it.
- line 790: what is meant by the "amplitude" of the TA-MSD? Unless the motion is confined (which is not considered here), the curves grow without bound.
- line 806: what is meant by "A mention apart..." This is likely an English usage problem, but I do not understand what the authors mean by this phrase.
- in Algorithm 1, the sampling of the displacement appears to be a scalar. For higher dimensional motion, is it assumed the different directions are independent?
- In Algorithm 3, more details are needed on how σ/γ are generated. It simply says "generate randomly". What underlying distribution is used?
- In Algorithm 4, I do not understand what the notation $(0,1,\dots,T)^{\alpha}$ means.
- Only at line 871 was it clear to me that the previous material was all just about 1D. The model used does not allow for correlations between the different directions.
- Line 896: assigning a probability mass function to the set of models but then just taking the most likely? What if the approach is essentially tied between two? It seems odd to require a PMF but then assign an all-or-nothing score as if a point estimate was given. This is addressed by the ROC metric, I suppose.
- Line 914: what is $\text{rank}_{\{\text{MAE},\text{F1},\text{RMSE}\}}$? The meaning of rank is undefined. Presumably it's the position in an ordered list based on performance in that metric but it is not clear.
- Line 961: What is the definition of a "matching changepoint" Does it need to be an exact time point or is there a window?

SUPPLEMENTAL 1:

- I realize it may not be feasible at this time, but I think have each time provide an optional (?) graphic for their approach would be helpful. This is especially true for ML methods where that visual can quickly and effectively convey the structure of the network that was used.

Point-by-point reply to reviewers of *Objective comparison of methods to decode anomalous diffusion* by G. Muñoz *et al.*

Reply to Reviewer 1

Comment R1.1 *This manuscript describes the results of a competition to compare methods for classifying and characterizing various forms of anomalous diffusion. It follows in the footsteps of previous community-wide competitions, such as the particle tracking challenge (Nat. Methods 2014) and the cell tracking challenge (Nat. Methods 2017). These competitions are very helpful to the community, as they give the reader a good overview of what methods are available and how the methods perform, as judged by criteria applied identically to all methods. Importantly, each method is applied by its own developers/protagonists, giving it the best chance to succeed, thus eliminating concerns about potential suboptimal use due to lack of familiarity or incomplete understanding.*

This challenge had good participation, with 15 teams applying different methods, some using machine learning, others using classical statistics, and some using a combination of the two. The challenge assessed the various methods' performance in three tasks: identifying the anomalous diffusion exponent α , determining the underlying diffusion model (choice of 5 different models), and detecting switches in diffusion properties within a single trajectory. These tasks were performed in 1D, 2D and 3D. Each participating team submitted results for whichever tasks and dimensions of interest/relevance to them.

The manuscript is overall well written, and I appreciated its combination of results presentation and detailed discussion of the different anomalous diffusion models, almost as a review/tutorial.

Reply to R1.1 We thank the reviewer for the endorsement and appreciation of this work.

Comment R1.2 *From my reading, the challenge resulted in quite a clear conclusion of which algorithms perform best, namely those based on machine-learning. This brings me to a critique of the abstract, where the conclusion is more open/vague. It is true, as the abstract says, that no single method performed best across all tasks. However, a group of methods did, and those were all based on machine learning. I believe this is a conclusion worth mentioning upfront in the abstract, so that a reader actually learns something from it. If other methods based on machine learning did not perform well, it is also worth discussing the difference between them and the more successful machine learning based methods. The purpose of these competitions is to guide both users and developers in terms of where to put their effort and energy, but this guidance can only happen if conclusions are stated clearly.*

Reply to R1.2 We thank the reviewer for this comment: We completely agree with the reviewer. Thus, we have now modified the abstract and the discussion to clearly state this conclusion about the results of the competition. The abstract now reads:

“Although no single method performed best across all scenarios, the competition revealed clear differences between the various approaches. In particular, approaches based on machine learning showed superior performance for all tasks.”

In the discussion (lines 502-509), we now state:

“For each dimension, we could identify a group of methods with comparable performance that greatly improve the precision of the anomalous diffusion exponent with respect to the baseline provided by the classical estimation of the MSD. These approaches were all based on machine learning, so we can infer that machine-learning-based methods can go beyond

classical statistics, probably because they can extract from the trajectories of complex models some information that is not easily assessed by classical statistics.”

Comment R1.3 *Task 1 (identifying α) and Task 2 (determining diffusion model) are well defined and the algorithms’ performance is well characterized. Task 3 (trajectory segmentation in the case of switches) however is lacking an important aspect, namely the question of whether a changepoint is detected in the first place. The task as currently stated is concerned with the accuracy of determining the switching point, but it is not clear what happens if no switch point is detected at all. These are two separate aspects of this task that should be analyzed independently. First, whether a switch point is detected at all (in fact, one can ask about sensitivity (whether the real switch point is detected) and specificity (whether many false positive switch points are detected)). Second, for trajectories where the switch point is detected, one can then ask the question that is currently addressed, namely the accuracy of the switch point location. It is possible that with their current analysis these two questions are convoluted into one performance measure, but this may produce misleading results.*

Reply to R1.3 The reviewer raises a very important point and we agree that it deserves further discussion. We simulated all trajectories having a changepoint. Trajectories with changepoints located at the endpoints were interpreted as not having an “actual” changepoint. Thus, as the reviewer says, our analysis of RMSE simultaneously considers changepoint localization precision, false positives, and recall. These choices in the design of the task limit the independent interpretation of these aspects. Nevertheless, we have now analyzed these aspects as far as possible and extended the relative discussion in the revised manuscript. We have therefore included a plot of the recall (i.e., the sensitivity) vs the false positive rate (Fig. 4d in the new version of the manuscript) and its discussion, as well as a plot of the Jaccard similarity coefficient (JSC) vs $RMSE_{TP}$ (also see next comment). The related text has been added to the section “Task 3: Segmentation of the trajectory”, lines 344-384. It reads:

“For the challenge purposes, we simulated all trajectories as having a changepoint that could be located at any position, including the endpoints. In this view, the presence of a changepoint at one extreme was interpreted as a trajectory not having an “actual” changepoint. Similarly, participants were required to always provide a prediction for the changepoint position. In the case of not detecting a changepoint, the predicted position should have coincided with one of the trajectory endpoints. With this design, the RMSE simultaneously provides an evaluation of the localization precision as well as of its specificity. We also performed further analyses to independently assess the sensitivity and specificity of the participating methods and gain further insight into their performance. Since Fig. 4a-c show that it is challenging to estimate the changepoint when it is located very close to the trajectory start/end points, we considered trajectories with a changepoint within $\epsilon = 20$ points from the start/end as not having a changepoint. The same criterion was applied to the predictions provided by each method. Predictions/ground truth pairs located at $\epsilon < t < L - \epsilon$ were counted as true positives. Predictions/ground truth pairs located at $t \leq \epsilon$ or $t \geq L - \epsilon$ were counted as true negatives. Mixed cases were considered as false positive or false negatives. Based on this classification, we could evaluate the recall (Equation (14)), the false positive rate (Equation (15)), and the Jaccard similarity coefficient (Equation (16)). We also calculated the $RMSE_{TP}$, defined as the RMSE obtained only for true positives.

The plot of the recall vs the false positive rate (Fig. 4d) shows that all submitted methods detect more than 92% of the inner changepoints but present a rate of false positives larger than $\approx 10\%$ and sometimes as high as $\approx 40\%$. We think that several factors might interplay to produce this behavior. As explained earlier, participants always provided a prediction for the changepoint position, the latter being equal to one of the trajectory endpoints if no changepoint

was detected. In the latter analysis, our distance-based criterion relaxes this requirement to a distance $\epsilon = 20$ points from the endpoints. Thus, the high false positive rate reflects the methods' limitations when dealing with changepoints close to the trajectory endpoints that, instead of being associated to no changepoint, are generally predicted to be more internal. Nevertheless, since the challenge metric does not explicitly account for false positive identifications, predicting an inner changepoint even when odds of predicting a false positive are high might be a conservative choice to keep the RMSE low. In some case, this effect is produced by the choice of the architecture. For example, in 1D and 3D, team E applied a strategy based on the averaging of predictions obtained through different networks. In this way, they could reduce the RMSE even for changepoints close to the trajectory endpoints (Fig. 4a), but it also led to a high rate of false positives (Fig. 4d-e), associated with contrasting predictions of the networks (e.g., a very early changepoint and a very late changepoint), averaging into an internal point.”

Comment R1.4 *The analysis of recall vs. $RMSE_{TP}$ in Fig. 4d needs better explanation.*

Reply to R1.4 We have now substituted the analysis of recall vs. $RMSE_{TP}$ with a plot of the Jaccard similarity coefficient (JSC) for binary classification as a function of the $RMSE_{TP}$. In fact, with respect to the recall, the Jaccard similarity coefficient also accounts for false positive detections. Therefore, this is a more representative metrics to display the relation between overall detection performance and precision (Fig. 4e in the new version of the manuscript). As suggested by the reviewer, we have also extended the explanation of this analysis in the section “Task 3: Segmentation of the trajectory”, lines 385-399. It reads:

“In addition, we aimed at exploring the relationship between the overall detection performance and changepoint localization precision. As a measure of detection performance, we used the Jaccard similarity coefficient for binary classification (Eqn.16) that, with respect to the recall, further accounts for false positive detection. The localization precision was instead estimated by the $RMSE_{TP}$ resulting from true positive identifications. The plot of the Jaccard similarity coefficient vs the $RMSE_{TP}$ (Fig. 4e) shows that, despite the false positive rate, all submitted methods show good overall detection performance and comparable precision ($RMSE_{TP} = 10-20$ points). Interestingly, performance of teams B and J improves with the dimensionality of the problem, consistently with the increase of information provided by the additional components of the motion. Team E also shows an improvement from 1D to 2D, in agreement with this explanation. The degradation of performance of team E in 3D can be ascribed to their approach to the problem through the independent training of three 1D networks, showing obvious limitations when applied to a diffusion model that is not the simple composition of 1D diffusion along orthogonal directions.”

Comment R1.5 *Also regarding Task 3, I find the results in Fig. 4e and f rather surprising. For both, I would have expected the accuracy of switch point detection to be worst along the diagonal, because the trajectory would not really be switching its diffusion model/degree of anomalous diffusion. In addition, also for both, I would have expected some symmetry in these matrices in the off-diagonal. For example, in Fig. 4e, how come switching from $\alpha > 1.5$ to $\alpha < 0.5$ has an RMSE of 23.9, while switching from $\alpha < 0.5$ to $\alpha > 1.5$ (just the opposite of the previous switch) has a much worse RMSE of 36.7? As another example, how can switching from $\alpha < 0.5$ to $\alpha > 1.5$ have a worse RMSE than switching from $\alpha > 1.5$ to $\alpha > 1.5$ (36.7 vs. 29.3)? The authors should investigate these results further, and discuss them in greater*

depth. There is some mention of asymmetry in the discussion (p15, lines 427-430), but that is too brief and does not even refer to Fig. 4e and f.

Reply to R1.5 We agree with the reviewer that the results presented in Fig. 4e and f are not straightforward to interpret. By design, trajectories simulated for task 3 present a change at least in the diffusion model or in the value of α . However, not all the models can have any value of α . Therefore, the plots in former Fig. 4e,f entangled the effects of anomalous exponent and α , plus the effect of noise, making difficult to extract the contribution of each. To highlight performance as a function of α and the diffusion model, we have now restricted this analysis to a single noise level. As it can be seen in the new figures Fig. 4f,g, the analysis of this subset clearly shows the expected behavior. We have modified the text in section “Task 3: Segmentation of the trajectory”, lines 400-415, as follows:

“The combination of α -exponents and diffusion models of the two segments is also expected to affect the changepoint localization precision. However, our dataset has a rich parameter space entangling several variables (anomalous model, α , noise, changepoint location) and some imbalance since not all the models can have any value of α . To highlight changes in RMSE due to a switch in α or in the diffusion model, we restricted the analysis to a subset of trajectories with a single noise level (SNR=10, Fig. 4f,g). Unsurprisingly, the RMSE is minimal when there is a large change in α , as between nearly immobile motion ($\alpha < 0.5$) to either superdiffusion ($1 \leq \alpha < 1.5$) or directed or ballistic motion ($1.5 \leq \alpha \leq 2.0$) (Fig. 4f). The worst case scenario is instead observed when both segments undergo mild sub- ($0.5 \leq \alpha < 1$) or superdiffusion ($1 \leq \alpha < 1.5$). The matrix shows a reasonable level of symmetry, considering the large heterogeneity of the dataset. However, in the presence of small changes of α , such as between $0.05 \leq \alpha < 0.5$ and $0.5 \leq \alpha < 1.0$, or between $1.0 \leq \alpha < 1.5$ and $1.5 \leq \alpha \leq 2.0$, the methods seem to detect changes involving an increase of α with better precision.

This dependence is related in a nontrivial fashion to the change in RMSE observed as a function of diffusion models (Fig. 4g). In fact, while FBM and SBM allow Brownian, sub- and superdiffusion, CTRW and ATTM do not allow superdiffusion, and LW does not allow subdiffusion. Changepoints associated with a switch of α but with no change of model are the most difficult to precisely locate. The smallest RMSE is observed when LW switches to CTRW. In contrast, models involving an abrupt (ATTM) or smooth change of diffusivity (SBM) are the most difficult to distinguish from the others.”

Regarding the mention of asymmetry in the discussion, we would like to point out that it does not refer to Fig. 4e,f but rather to the curves in Fig. 4a-c, that we relate to the non-stationarity of the models. To further clarify this point, the text in the discussion (lines 548-559) has been modified as follows:

“Yet, the submitted results showed an interesting asymmetry: The changepoint localization precision seems not only to depend on the relative length of the segments but also on the changepoint location (Fig. 4a), producing a lower RMSE for changepoints located at the beginning of the trajectory. Similarly, the methods show best performance in detecting α and diffusion model for the first segment (Fig. 4b-c). We believe that this is at least partly a consequence of the inaccurate detection of the changepoint and the non-stationarity of some models. The inexact localization of the changepoint produces two spurious segments, altering the tail of the first segment and the initial point of the second by removing or adding spurious points. For non-stationary models, the initial point encloses information about the initiation of the physical process, thus improper segmentation impacts more severely the evaluation of the second segment [58].”

Comment R1.6 *In terms of performance measures, why do the authors use MAE to compare predicted and ground truth alpha, but then use RMSE to compare the predicted and ground truth changepoint position? Both are comparing individual values to each other, and it is not clear why this difference in treatment, which adds yet another acronym for the reader to remember while reading the manuscript.*

Reply to R1.6 We agree with the reviewer that both MAE and RMSE express errors on average model predictions and might both be used to assess performance of regression methods, such as those in Task 1 and 3. For task 1, we opted for using a linear score such as MAE due to the narrow range of possible values of $\alpha \in [0, 2]$. However, for the assessment of the changepoint position in task 3, we believe it to be more appropriate to use a quadratic metric that gives a relatively high weight to large errors and penalizes methods providing a prediction far away from the ground truth. We have now stated this choice in the manuscript, by adding the following text to the section “Methods. Metrics. Challenge metrics”, when introducing the RMSE (lines 1099-1101):

“Unlike for T1, where we used the MAE, in this case we opted for the RMSE. This quadratic metric gives a higher weight to large errors, thus penalizing methods that provide predictions very far from the ground truth.”

Comments on figures:

Comment R1.7 *As a major critique of manuscript organization/data presentation, I believe that most applications of anomalous diffusion analysis are for 2D trajectories. For example, among the experimental data presented in the paper, 3 out of the 4 datasets are in 2D. Thus, I think it will be most useful to present the 2D results in the main figures, instead of the 1D results currently presented. This applies to Fig. 2d-g, Fig. 3d-g and Fig. 4 (all panels I think).*

Reply to R1.7 We agree with the reviewer that several experiments, in particular in biology, provide 2D trajectories or 2D projections of 3D motion. However, our rationale was to first present results for 1D in the main manuscript and then show those for 2D and 3D in the supplementary material. This choice is motivated by the fact that 1D is conceptually easier to understand, permitting us to focus the reader’s attention on the details of the anomalous diffusion models and the implemented methods. This also permits us to make a more pedagogical presentation of the results. In fact, we also follow the same organization when describing the physical models and their simulation. In principle, all 1D methods can be used for larger dimension component-wise. Many methods and models are first developed in 1D and then extended to higher dimensions. In fact, several 2D methods participating to the challenge are just the extension of the corresponding 1D version to two components (even though this is not necessarily the optimal strategy). Also, we believe this is the reason why we obtained a larger participation to the 1D tasks. Therefore, this choice also allows us to assess the performance of a larger set of methods that might eventually be extended to higher dimensions.

We have now clarified this in the introduction (lines 157-164):

“Although several experiments provide 2D and 3D trajectories, we first present and discuss in detail the results obtained for the 1D trajectories. This choice is driven by the fact that the 1D-case is conceptually easier to understand, thus complex methods are in general first developed in 1D and then extended to multidimensional space, as testified by the larger participation for this dimension. Thus, it allows us to assess the performance of a larger set of methods including those that might eventually be extended to 2D and 3D. We follow the same rationale when describing the physical models and their simulation.”

Comment R1.8 *In Fig. 2a-c and Fig. 3a-c, it is not clear what the different color bars represent. The legend in Fig. 2 says “colors represent mean absolute error contributions for each underlying diffusion coefficient.” But then what does it mean to stack them up? Should they not be averaged to get overall MAE? For example, what do the yellow bars in Fig. 2a mean? Does this mean the MAE for SBM is 0.3 for say Team K, or is this just an artifact of stacking the bars, and their real MAE is actually j 0.1?*

Reply to R1.8 We apologize for the lack of clarity. The total length of the bar (sum of all colors) represents the value of the metric calculated over the whole dataset, as reported in the x -axis. The colored parts of the bars account for the relative value of the metric calculated on specific subsets and normalized such that the sum of all colored bars gives the overall metric value. Figure legends of Figs.2-3 have been amended to clarify this point by adding the following text:

“The colors represent the relative contributions to the overall mean absolute error (MAE) calculated for each underlying diffusion model and normalized such that the sum of all contributions gives the value of the same metric calculated over the whole dataset.”

and

“The colors represent the relative contributions to the overall F_1 -score calculated for different ranges of anomalous diffusion exponents and normalized such that the sum of all contributions gives the value of the same metric calculated over the whole dataset.”

Comment R1.9 *As far as I understand, the confusion matrix in Fig. 3e should be identical to the confusion matrix in Suppl. Fig. S11 for Team E (first confusion matrix in Suppl. Fig. S11). But they are different. Why?*

Reply to R1.9 We apologize for an error leading to this comment. Because of a code typo, only numbers in the confusion matrix of Fig. 3e were transposed. We have now corrected the figure, which indeed looks identical to the confusion matrix of Suppl. Fig. S11 for Team E.

Comment R1.10 *Overall, some of the confusion matrices in Figs. S11-S13 add up to 100 per row and per column, and some do not. Shouldn't all of them be adding up to 100 per row and per column?*

Reply to R1.10 Confusion matrices can be normalized according to different criteria, i.e., based on rows, columns, or total. In our case, we used column-based normalization, so they should add up to 100 per columns (apart for numerical rounding), but not necessarily per rows. More in detail, numbers in matrix cells represent the number of correctly and incorrectly classified trajectories for each ground-truth model as percentages of the number of observations of the corresponding ground-truth model. Thus, the percentages of correctly classified trajectories can be thought of as class-wise recalls. Still, because of rounding, sometimes sums don't give exactly 100. The description of the used normalization and the meaning of the numbers has been added in the figure captions of Fig. 3e and Figs. S11-S13:

“Numbers in matrix cells represent the number of correctly and incorrectly classified trajectories for each ground-truth model as percentages of the number of trajectories of the corresponding ground-truth model (column-based normalization, so that their sum along the columns should add up to 100, with minor deviation due to rounding). Thus, the percentages of correctly classified observations can be thought of as class-wise recalls.”

Other comments:

Comment R1.11 p. 7: *What exactly does micro-average refer to in the F1-score?*

Reply to R1.11 For multi-class classification problems, micro-averaging and macro-averaging are methods for evaluating scoring metrics such as precision, recall, and F1-score. A macro-average computes the metric independently for each class and then takes the average (hence giving all classes the same weight), whereas a micro-average will aggregate the contributions of all classes to compute the average metric. Micro-average is generally preferable when class imbalance is present. The description of micro-average for the case of the F1-score has been added in Methods (lines 1081-1092):

“For multi-class classification problems, scoring metrics such as precision, recall, and F₁-score can be computed as a macro-average (which evaluates the metric independently for each class and then take the average, giving all classes the same weight), or as a micro-average (which aggregates the contributions of all classes to compute the average metric). Micro-averaging is generally preferable when class imbalance is present. Although the challenge was based on a balanced dataset with each class equally represented, we used a micro-averaged F₁-score in order not to provide any hint to participants about the content of the dataset. The micro-averaged F₁-score was calculated as

$$F_1 = \frac{2TP}{2TP + FP + FN}, \quad (1)$$

where TP, FP, and FN represent true positives, false positives, and false negatives calculated over the whole dataset, respectively.”

and a reference to the equation has been added in the main text (line 204).

Comment R1.12 p.7: *The authors should refer to Eq. 12 in Materials and Methods section when mentioning “mean reciprocal rank.”*

Reply to R1.12 We thank the reviewer for this suggestion, the reference has been added.

Comment R1.13 p. 8 *“Participating teams and methods”*: *When mentioning the grouping of approaches, it would help to explicitly list the different teams belonging to each approach.*

Reply to R1.13 Following this comment and a suggestion of reviewer 3, we have restructured this section and added a reference to Table 1, where all teams are listed and classified according to the type of model, input data, a and length-specificity. The sections now reads:

“Among the participants, we could distinguish fifteen substantially different approaches (Table 1 and Supplementary Note 1). We classify the approaches based on three different criteria, as detailed in Table 1. First, we group methods based on the type of approach used, whether involving machine-learning or classical statistics. A large majority of methods are based on machine-learning architectures, such as recurrent neural networks (RNN), convolutional neural networks (CNN), gradient boosting machines, graph neural networks, extreme learning machine (ELM), or sequence learners. Other methods are based on statistical approaches, such as Bayesian inference, temporal scaling, and random interval spectral ensemble (RISE). A second grouping involves the type of input data used. Some methods employed feature engineering using classical statistics as an input, whereas other were simply fed raw trajectories. A

further classification is based on whether methods required a specific training or model for different (ranges of) trajectory lengths (length-specific) or not. Several methods could be directly used or easily adapted to run multiple tasks. ”

Comment R1.14 *p. 10, line 261: Clarify what is meant by “smoothly” in the phrase “performance varied more smoothly.”*

Reply to R1.14 The text has been modified to clarify what we meant, it now reads:

“For each dimension of this task, a different number of methods showed comparable performance (Fig. 3a-c). For each dimension, we selected the 2 teams that achieved top scores.”

Comment R1.15 *p. 12: Please mention dimensionality of experimental data trajectories. Yes this is mentioned in Supplementary Note 2, but that is far away in the manuscript and I don’t think there is any reference to it from the main text.*

Reply to R1.15 Following the reviewer’s suggestion, the trajectory dimension has been added when first mentioning the experiments.

Comment R1.16 *p. 14, lines 393-394: The authors state that providing precise and unbiased prediction for short and noisy trajectories is important for a method’s performance. But I am not sure how the authors conclude this from their data and analyses. From the figures, all methods are worse for shorter trajectories and better for longer trajectories. And all methods are worse for lower SNR and better for higher SNR. While this statement is kind of self-evident, I am not sure how it stems from the data and analyses of the manuscript, as implied by that statement.*

Reply to R1.16 The reviewer is correct, the statement is not fully proven by the plots in Figs. 2 and 3. Therefore, we have rephrased the statement and added a Supplementary figure (Fig. S29) showing the correlation between metrics obtained over a subset of short and noisy trajectories with respect to the same metrics obtained over the whole dataset. The new figure has been referenced in the discussion:

“Top-ranking methods provide precise and unbiased predictions also for short and noisy trajectories, as shown by the correlation between metrics calculated over a subset of trajectories ($L < 200$, $SNR= 1$) with respect to the same metrics obtained over the whole dataset (Supplementary Fig. 29). This is a major improvement for trajectory analysis, since it enables collecting information from short and noisy tracks (e.g., those obtained by SPT PALM [?]) and from time segments of trajectories exhibiting heterogeneous behavior, without further averaging. However, the aspect that mostly boosts the overall performance is the ability to extract the anomalous diffusion exponent (an intrinsic ensemble property) for non-ergodic models from single trajectories (Fig. 2d).”

Comment R1.17 *p. 34, line 755: Reference to a non-existent Fig. 1d.*

Reply to R1.17 We apologize for the typo, the text has been corrected, it now reads “Fig. 1c”.

Comment R1.18 p. 35, line 788: *The authors state that “the EA-MSD does not converge to TA-MSD in the infinite time limit.” Should the statement not be the other way around, namely “the TA-MSD does not converge to EA-MSD in the infinite time limit.” As the authors themselves state on p. 4, “Traditionally, the MSD is defined as the ensemble average over a group of tracers.”*

Reply to R1.18 The reviewer is right. We thank the reviewer for identifying this error in the text. We have now amended the text.

Comment R1.19 p. 35, line 789: *This statement needs a reference.*

Reply to R1.19 We have now added a reference to Metzler et al, Phys.Chem.Chem.Phys., 2014, 16, 24128.

Comment R1.20 p. 35, line 790: *What is the “amplitude” of the TA-MSD?*

Reply to R1.20 The text has been changed, it now reads:

“the value of the TA-MSD for different trajectories at a given timelag ...”

Comment R1.21 p. 44, line 961: *Please define “Matching changepoints.” Is there a tolerance, or is the match exact?*

Reply to R1.21 The section including this information has been largely modified to provide a clearer explanation of the procedure:

“For the assessment of the changepoint localization error in T3, we followed two different evaluation approaches. For the challenge evaluation, we simply quantified the RMSE. Trajectories showing no changepoint were considered as having a dummy changepoint either at index 1 or 199. However, to get a better understanding of methods’ performance, we also considered an alternative analysis. For this, trajectories with ground truth and predicted changepoints within a distance $\epsilon = 20$ from the start/end points were considered as not having a changepoint. We thus considered four cases:

- predicted and ground-truth positions located at $\epsilon < t < L - \epsilon$, counted as true positives (TP);*
- predicted and ground-truth positions located at $t \leq \epsilon$ or $t \geq L - \epsilon$, counted as true negatives (TN);*
- the predicted position located at $\epsilon < t < L - \epsilon$ but the ground-truth located at either $t \leq \epsilon$ or $t \geq L - \epsilon$, counted as false positive (FP);*
- the predicted position located at either $t \leq \epsilon$ or $t \geq L - \epsilon$. but the ground-truth located at $\epsilon < t < L - \epsilon$, counted as false negative (FN).*

Based on this classification, we evaluated the recall (also known as sensitivity):

$$\text{recall} = \frac{\text{TP}}{\text{TP} + \text{FN}}; \quad (2)$$

the false positive rate:

$$\text{FPR} = \frac{\text{FP}}{\text{FP} + \text{TN}}; \quad (3)$$

and the Jaccard similarity coefficient (JSC) for binary classification:

$$\text{JSC} = \frac{\text{TP}}{\text{TP} + \text{FP} + \text{FN}}. \quad (4)$$

We also calculated the RMSE_{TP} , corresponding to the RMSE obtained only for prediction/ground-truth pairs classified as true positives. ”

Comment R1.22 p. 45, line 980: The authors state “As shown in Fig. 2, such a simple fit produces results comparable with the best methods for ergodic models with high SNR and long trajectories.” Where is this shown in Fig. 2?

Reply to R1.22 The reviewer is correct, the previous statement was not fully substantiated by the plots in Fig. 2. Therefore, the text has been modified to be consistent with the results shown in Fig. 2. It now reads:

“As shown in Fig. 2d, for ergodic models (FBM), such a simple fit produces results comparable with the best methods.”

In addition, to allow the reader to take advantage of the rich parameter space explored by our datasets, we have created an interactive tool that allows the user to display results of the challenge for selected subsets (available at <http://andi-challenge.org/interactive-tool/>). An example is provided in Fig. R1, reporting the ranking of methods for Task 1 in 1D for FBM only at SNR=10, for FMB only at SNR=1, and for all models at SNR=10. To further stress this possibility, we have also added the following sentence (lines 1188-1191):

“In addition, as it can be observed using the interactive tool (<http://andi-challenge.org/interactive-tool/>), the estimation of α through the fit of the TA-MSD even outscores the other methods for ergodic models (FBM) at the highest SNR (e.g., SNR= 10).”

Comment R1.23 p. 45, lines 983 – 995: This text seems to be there only for completeness. This should be stated explicitly.

Reply to R1.23 We have amended the text by adding the following sentence:

“For the sake of completeness, we would like to mention other statistical approaches not considered in the challenge that can be used to tackle T1.”

Comment R1.24 p. 46, Eq. 14: This equation needs either derivation or a reference.

Reply to R1.24 We have added the following text:

“The random predictor $\text{RMSE}_{\text{random}}$ can thus be calculated as the RMSE for a trajectory with a changepoint at position t_{GT} and random predictions t of the changepoint drawn from a uniform distribution in the range $[0, L]$

$$\text{RMSE}_{\text{random}}(t_{\text{GT}}) = \sqrt{\frac{1}{L} \int_0^L (t - t_{\text{GT}})^2 dt} = \sqrt{\frac{t_{\text{GT}}^3 + (L - t_{\text{GT}})^3}{3L}}, \quad (5)$$

where L is the trajectory length.”

Figure R1: Metrics obtained for methods participating in Task 1-1D for FBM only at SNR=10 (top panel), for FBM only at SNR=1 (middle panel), and for all models at SNR=10 (bottom panel).

Comment R1.25 *Supplementary Note 1: The descriptions of the different algorithms are full of acronyms. Please define/spell out all new terms. Also, some methods state a reference, while others do not. This might be because the methods without a reference were developed for the purposes of the challenge. In this case, they should still state a reference, and it can say something like “this work.” This eliminates any ambiguity.*

Reply to R1.25 The acronyms have been spelled out. Following the reviewer’s suggestion, we have also added a reference to each method in order to avoid any ambiguity.

Reply to Reviewer 2

Remarks to the Authors:

Comment R2.1 *Gorka Munoz-Gil et al., present a summary of how effective is the performance of different data analysis methods applied to stochastic trajectories, in particular those involving anomalous diffusion. The different analysis methods are from typical statistics (Bayesian inference) to machine learning. A competition called AnDi, was open to the community, and based on this one can gain new insights on the different methods, for example which performs better.*

Trajectory data gained these days, from single molecule tracking to other fields like finance is clearly enormous. In many cases such paths are modelled with anomalous diffusion deviating from traditional Brownian motion. Usually, it is easy to tell from experimental data that Einstein's theory of Brownian motion fails. However, predicting the type of non-Brownian dynamics and even determining the exponent alpha, characterizing the mean square displacement, can be hard. Hence, I believe the authors have served a wide range of communities, by making this type of analysis through competition. I am not familiar with a similar scientific battle and hence the whole approach is timely and refreshing. In short, it is my opinion that the authors promoted the field of data-analysis relevant to many fields.

The authors define three tasks:

- *Task 1 determine the anomalous diffusion exponent alpha.*
- *Task 2 determine the underlying diffusion model.*
- *Task 3, does the diffusion process change within the observation window (trajectory segmentation).*

They then check which method performs better (see more details on the methods in the manuscript). I believe the definition of tasks are useful and reasonable. For example the value of alpha determines how fast certain molecules diffuse in the cell, and this is clearly important for bio-chemical reactions and in general for the ticking of the cell.

The authors show clearly how effective the different methods are for succeeding in the three different tasks, involving noise corrupted simulation data with predefined anomalous exponents and specific diffusion /transport models. Particularly, the authors show how machine learning methods are better for determining the anomalous exponent in the data set from simulations (Task 1), even for short time series. For determining which diffusion model the simulation data is described by (Task 2) for a broad range of values of alpha. And Finally, for discriminating between the alpha exponent and diffusive models in time series cut randomly by half. In all of them Machine learning methods exhibit a better performance.

Plus the authors show the performance of the best competitors for solving Task 1 and Task 2 on experimental data sets for four different experiments. (diffusion of mRNA inside the cell, telomeres inside the nucleus, plasma receptors on the membrane and atoms in optical lattices). The majority of them were consistent with the values and models reported in the literature, just the second and last case shows discrepancies in the value of the anomalous exponent. More on this see below.

Good features:

Comment R2.2 *The paper is interesting and clearly exhibits how machine learning methods give better results if compared with classical statistical methods when time series with anomalous diffusion are studied. The comparison between different methods is insightful.*

Reply to R2.2 We thank the reviewer for the appreciation of our work and for his/her insightful comments.

Some questions/remarks:

Comment R2.3 *The authors don't explain why Task 3 was not incorporated in the study of experimental data sets.*

Reply to R2.3 The reason is that, unfortunately, we do not have an experimental dataset consisting of trajectories featuring changes of diffusion models and/or anomalous diffusion with availability of previous analysis for comparison. Still, we applied the submitted methods to the same experimental trajectories discussed for Tasks 1 and 2. The methods did not evidence a significant presence of changepoint in these data, as one should expect. We have now added a paragraph at the end of the section “Analysis of experimental data” (lines 493-497) to explain this point:

“The methods participating in T3 were not initially planned to be applied to the analysis of experimental data, due to the lack of trajectories featuring changes of diffusion models and/or anomalous diffusion exponent with availability of previous analysis for comparison. However, when applied to some of the experimental trajectories described above, they did not evidence a significant occurrence of changepoints, as expected.”

Comment R2.4 *Having a comparison between machine learning segmentation and for example local convex hull method could be useful.*

Reply to R2.4 Besides the fit of the MSD, which is a traditionally well established method and was applied by the organizers, we decided to let developers use their own tools for the challenge. As also pointed out by reviewer 1, this choice prevents concerns about potential suboptimal use of the methods. Although, as pointed out in the introduction, the convex hull method has been applied to similar problems (FBM with different α 's, Lanoiselée and Grebenkov, Phys. Rev. E 96, 022144 (2017)), no team participating in the challenge developed an approach based on the same principle. We have now clarified our choice in the section “Competition design” (lines 176-180):

“For the competition, we let developers build and use their own tools to provide predictions for the common dataset. This choice might have somehow limited the participation and comparative assessment of methods based on popular approaches. However, it was meant to prevent drawbacks associated with the suboptimal use of methods.”

Comment R2.5 *Did we learn something new on the systems involved? for me a surprising result was the analysis of the motion of atoms in one dimensional optical potentials. Based on theoretical arguments it was suggested previously that the process belongs to the CTRW class of the normal type. The microscopical model in mind was an underdamped particle in a periodic structure, a perfectly normal model with $\alpha=1$. Now the authors show, based on available experimental data, that the system exhibits sub-diffusion (though errors are large, see fig. 5h). One could argue that this is a nice demonstration of the power of the methods involved. However, at a closer look we may suspect that the conclusion is simply misleading. To see this recall that the training session did not include the stochastic model which was suggested in the literature. See letter titled CTRW for a particle in a periodic potential PRL, 123, 070602 (2019) in particular please pay attention to the under-damped Langevin model (Eq. 2 there). Note that the authors (Dechant et al.) in mentioned PRL, find excellent agreement between theory and experiment without any fitting parameter, which is most likely not a coincidence. Hence, I think authors of the present paper are making a daring claim. To solve this mystery I suggest the authors meet the following challenge. Please train the machine with the model in the mentioned PRL, and see what is the verdict with respect to the experimental data. I would not be surprised if the Langevin model, in the presence of periodic potential, works fine. If so we*

may reach the conclusion that depending on the input given for the purpose of machine learning, we get vastly different pictures of data (however for now this is merely my speculation). Any way, it is hard for me to accept that a model without fitting parameters is simply wrong, and unfortunately I am in a state of confusion. As mentioned, I cannot say if this is a limitation of the previous analysis or the present one, and the authors can clarify this point by adding the requested model to the tool box.

Reply to R2.5 The reviewer is right, the diffusion of the atoms in the periodic potential is very well explained by a CTRW where the parameters (dwell time and hopping distance) can be deduced from microscopic mechanisms, as described in the PRL article the reviewer mentions. However we would like to point out that: (i) the normal CTRW is indeed included in the training dataset; (ii) the results provided by the participant are compatible with the interpretation provided in the PRL article and no claim is made about a different model; (iii) it is obviously more precise to perform a fitting with an *ad hoc* model when possible, but this is beyond the scope of our work (and practically unfeasible since all teams would have to retrain and reoptimize their approaches).

Before commenting in more details on the interpretation of the results, we would like to further stress that the trajectories composing this dataset are extremely short (≈ 10 points). Based on results obtained from simulations, we know that for short trajectories the prediction error increases both when assessing the diffusion model and the anomalous diffusion exponent. However, since we have a system where the microscopic mechanisms are well known, we aimed at using it as a benchmark to check the predictive limits of the different approaches for very short trajectory length in a real scenario.

For the challenge, the experimental trajectories were independently analyzed by methods participating in Task 1 (inference of α) and Task 2 (model classification). In spite of the very challenging conditions, the best methods participating in Task 2 predicted CTRW to be the most likely underlying model for these experiments, in agreement with previous results. The fact that several approaches yield reasonable agreement is, in our opinion, a success of the challenge. Given the short trajectory length and the lack of any prior information, we find it even more remarkable. Please also notice that for several methods, this prediction is fully independent from the estimation of α . When attempting to estimate the anomalous diffusion exponent (Task 1), the methods for Task 1 produce a wide distribution of exponents. As shown in Fig. S24, several methods provide a median exponent very close to 1, as previously predicted. However, they show the tendency to underestimate α with respect to the previous results, predicting a slightly subdiffusive behavior. We do not know the reasons for this prediction. We might speculate that is likely due to the very short trajectories and/or, to the training dataset, e.g. not including trajectories with the same level of noise as in the experiments. However, the main point is that we do not make any daring claim about the physics of the system, mainly because it is not the objective of our study. Nonetheless, in the manuscript we do comment on this discrepancy, under the assumption that the previous predictions provided by the authors of the PRL article are correct. To clarify this point, we have rephrased and extended part of the discussion associate to these experiments (lines 472-492):

“To demonstrate the applicability of these methods beyond biological systems and at different spatio-temporal scales, we included a dataset with 1D trajectories obtained for single atoms moving in a 1D periodic potential and interacting with a near-resonant light field that acts as a thermal bath [50](Fig. 4d). These data were originally interpreted as evidence of CTRW with $\alpha = 1$ [50]. Subsequently, the CTRW was deduced from microscopic parameters reproducing the trajectories without free parameters [56]. Because of the intrinsic complexity of this experiment, the trajectories were extremely short (≈ 10 datapoints), a regime that challenges the predictive power of any approach. Indeed, in this range of trajectory length all the methods showed rather large uncertainties (Fig. 2f and Fig. 3f). However, since the microscopic mechanisms are well known, we aimed at using these experiments as a benchmark to check the predictive limits of the different approaches for very short trajectory length in a real scenario. The top regression methods for such short trajectories in 1D provided distributions spread over

a wide range of α with medians between 0.8 and 0.91 (Fig. 5h). The results of model classification were also less conclusive with respect to the previous cases, likely a consequence of having short trajectories and of having $\alpha \simeq 1$, a regime where detectable differences among models are reduced (as shown in Fig. 3d). Still, the CTRW was the most-likely model for 4 of the 5 top-scoring methods (between 28% and 48%, Fig. 5l), thanks to the capability of these methods to extract information from the microscopic dynamics of the generative models and not only from the long-term properties of the trajectory and its MSD.”

The reviewer is also correct that the training might play a role. It would be fully reasonable that a training using only CTRW with $\alpha = 1$ would yield better results. However, it would also add a bias as compared to the other models. Moreover, as mentioned above, this goes beyond the scope of this article, since our objective is to compare general methods that can provide an output with no prior information about the specific system.

Comment R2.6 *In general the AnDi was built for models that can exhibit either anomalous or normal diffusion (alpha is a control parameter). What happens when you add models which are perfectly normal, with no knob on alpha? This is not just a nasty question, real data that is perfectly normal might be wrongly interpreted as anomalous.*

Reply to R2.6 Normal models are included in the datasets for all challenge tasks. In fact, one of the objectives of the challenge was to objectively compare methods for the inference of the anomalous diffusion exponent (Task 1). All the theoretical models included in the challenge converge to normal diffusion for $\alpha = 1$, although with different microscopic features. The normal case was therefore analyzed for all diffusion models and associated results can be seen in Figs. 2e, S7-9. Trajectories simulated from these normal models can be interpreted as anomalous in the presence of large errors in the estimation of α . However, as it can be seen in Fig. 2e and Figs. S7-S9 the best methods show a distribution of predictions for the exponent α centered around the ground-truth value and quite narrow. We have further commented about this point in section “Task 1 - Inference of the anomalous diffusion exponent” (lines 268-271):

“... the best-in-class method achieves similar performance over the whole range of α , whereas TA-MSD has a lower accuracy for $\alpha \simeq 0.5$ to 1. Obtaining precise predictions for $\alpha \simeq 1$ is particularly relevant, since the correct assessment of the exponent in this regime would further allow the discrimination between normal and anomalous diffusion.”

Comment R2.7 *In other words, how important is the input of five anomalous diffusion models (ATTM, CTRW, FBM, LW, and SBM)?? why these models? with respect to simulation competition this is fine, as one needs to start with certain models. But when analysing real data, can we be sure that this choice of five models does not create a bias? for example over damped Langevin models in the presence of local fields is a reasonable model to start with. Why was it excluded?*

Reply to R2.7 As the reviewer says, the zoology of anomalous diffusion models is rather broad. We focused on these five (ATTM, CTRW, FBM, LW, and SBM) because they are general frameworks used to interpret and model several experimental datasets. We think this choice represents a reasonable trade off to ensure enough generality while limiting the number of models.

We fully agree that, when available, a parameter-free model designed ad hoc for a specific problem might offer better performance. In this fortunate cases, it might not even be necessary to apply the tools employed in the AnDi challenge. As detailed in the discussion, we are also aware of the limitations of the classification task, such as the lack of none-of-the-above choice. However, we believe that adding a

specific model to the training set goes beyond the scope of our work since it won't benefit the discussion about methods performance. However, we have further expanded the discussion of these issues in the discussion (lines 536-541):

“However, a limitation of all these classification approaches is that they can only choose among the diffusion models provided in the training. To robustly extend model classification to actual experiments, it can be useful to further widen the palette of models (e.g., by using ad hoc models), add a none-of-the-above class, and/or to include some metric of the confidence of the estimation (e.g., by using an entropy measure calculated on the predictions of an ensemble of machine-learning models).”

Comment R2.8 *To conclude, I find the paper very interesting and promising. Some clarification on the applications to real data is needed, while the competition shows clear insights (machine learning wins) with respect to simulated data.*

Reply to R2.8 We thank again the reviewer for appreciating our work and for the constructive comments, which we hope to have fully addressed.

Minor issues:

Comment R2.9 *on line 113 authors write that the use of TA MSD is not valid, for the determination the exponent alpha. While the intention of the authors is clear this sentence is too strong. Simply said we have two ways to determine alpha, either with the time or with the ensemble average, though neither can be considered superior with respect to the other. In the bulk of the paper the aim is to obtain the exponent of the EA MSD, which is fine, but seems to me arbitrary, in fact in the realm of single molecule tracking, I think the opposite is natural.*

Reply to R2.9 For the non-ergodic models considered in the AnDi challenge, the exponent α , besides describing the long-time behaviour of the EA-MSD, also characterizes the physical properties of the systems. For example, in CTRW, it controls the tail of the waiting time distribution. On the other end, for several of these models the TA-MSD shows a linear behavior (i.e., the value of α would always be 1), thus does not allow to infer crucial information about the system. We have modified the introduction (lines 111-119) to clarify this point, it now reads:

“... while for a stationary motion in a homogeneous medium EA-MSD and TA-MSD have the same exponent, several systems are intrinsically heterogeneous and non-stationary [19,20], which can lead to non-ergodicity (i.e., the non-equivalence of time and ensemble averages). Typically, the exponent α of the EA-MSD characterizes the physical properties of the systems (e.g., the trapping time distribution in CTRW or the time-dependence of diffusivity in SBM). However, in several non-ergodic systems, the TA-MSD shows a linear behavior with respect to the timelag in the long time limit even when $\alpha \neq 1$ [4]; ...”

Comment R2.10 *line 185 explanation on the meaning of micro-averages is needed.*

Reply to R2.10 For multi-class classification problem, micro-averaging and macro-averaging are methods for evaluating scoring metrics such as precision, recall, and F1-score. A macro-average computes the metric independently for each class and then take the average (hence giving all classes the same weight),

whereas a micro-average will aggregate the contributions of all classes to compute the average metric. Micro-average is generally preferable when class imbalance is present. The description of micro-average for the case of the F1-score has been added in Methods (lines 1081-1092):

“For multi-class classification problems, scoring metrics such as precision, recall, and F₁-score can be computed as a macro-average (which evaluates the metric independently for each class and then take the average, giving all classes the same weight), or as a micro-average (which aggregates the contributions of all classes to compute the average metric). Micro-averaging is generally preferable when class imbalance is present. Although the challenge was based on a balanced dataset with each class equally represented, we used a micro-averaged F₁-score in order not to provide any hint to participants about the content of the dataset. The micro-averaged F₁-score was calculated as

$$F_1 = \frac{2TP}{2TP + FP + FN}, \quad (6)$$

where TP, FP, and FN represent true positives, false positives, and false negatives calculated over the whole dataset, respectively.”

and a reference to the Equation has been added in the main text (line 204)).

Comment R2.11 *Trajectory segmentation, implies a sudden jump in the underlying control parameters, this is a special case of the more general time dependent change of the process (the latter category is of course endlessly large). I am not sure if any of the methods could deal with such a case.*

Reply to R2.11 The methods taking part in task 3 were specifically designed for the challenge and have not been tested on processes with a time dependent change of diffusion model or α . We have added a sentence to the section “Task 3: Segmentation of the trajectory” (lines 331-333) to clarify this point:

“The methods taking part in T3 were specifically designed for the challenge and have not been tested on other time-dependent processes, e.g., such as those involving a continuous change of anomalous diffusion properties.”

Comment R2.12 *Power spectrum analysis, convex hull methods, p-variation approach, were not used and not mentioned (unless I missed this). Is there a deep reason for this? or is this simply because these methods were not represented in the competition?*

Reply to R2.12 For the challenge, we decided to let developers use their own tools. As also pointed out by reviewer 1, this choice prevents concerns about potential suboptimal use of the methods. In addition, several methods previously proposed can only partly solve the tasks proposed by the challenge. Therefore, they have been mentioned in the introduction:

- *“Other statistical parameters have been proposed for this task and algorithms based on the combination of several estimators allow to distinguish between pairs of models [21–24], but there is no general consensus on how to unambiguously determine the underlying diffusion model from a trajectory”*

where Ref. 22 concerns the p-variation (lines 124-127).

- *“In recent years, advances in fluorescence techniques have greatly increased the availability of high-precision trajectories of single molecules in living systems [35], producing an increasing drive to develop methods for quantifying anomalous diffusion [16, 25, 32, 36–39]”*

where Ref. 38 refers to the power spectral analysis (lines 140-146).

- *“Only recently, attempts have been made to determine changepoints with respect to a switch in α [25, 32, 33] and diffusion model [34].”*

where Ref. 34 is about the application of the convex hull method to FBM with a switch in α (lines 136-139).

In addition, as described in the Supplementary Information, the methods developed by teams L and N used features extracted from the power spectrum analysis. However, no team participating in the challenge developed an approach uniquely based on these methods.

Reply to Reviewer 3

Remarks to the Author:

Comment R3.1 *This is a very interesting paper describing the results of an interesting challenge developed to test different approaches to analyzing anomalous diffusion. Overall, I think this is a strong contribution of significant merit and warrants acceptance into Nature Methods [ED: all comments also relevant for Nature Communications]. I do, however, have a few comments/concerns/questions. They are organized below by section.*

Reply to R3.1 We thank the reviewer for enthusiastically endorsing our work.

INTRODUCTION

Comment R3.2 - *I am very happy to see the list of issues with using the MSD. Despite this, the paper elevates the estimation of the anomalous coefficient in terms of that seen on the MSD to be on par with determining the underlying model (and, presumably, its parameters). Can the authors provide additional justification for the continued use of the anomalous coefficient and the MSD? Does it provide information that is useful to the practitioner or is it simply the culture to look at data this way?*

Reply to R3.2 Historically, the anomalous diffusion exponent (obtained from the fit of the MSD) has been used for detecting the occurrence of anomalous diffusion and provide a quantifiable parameter related to the diffusion mechanism, in addition to the diffusion constant. The simplicity of its calculation has produced the wide use (and sometimes abuse) of this parameter. However, besides being used as a discriminator of normal/anomalous diffusion, the exponent α is associated with relevant physical mechanisms, such as target finding and reaction rates. Notably, for several of the models included in the challenge and described in the Methods, the anomalous diffusion has a physical meaning that goes beyond the long-time scaling of the EA-MSD. For example, it controls the tail of the distribution of trapping times in CTRW, or the rate of change of the diffusivity in SBM. As such, its calculation allows gaining physical insight on the diffusion process. To further stress this point, we have extended the explanation given in the introduction (lines 111-119):

“... while for a stationary motion in a homogeneous medium EA-MSD and TA-MSD have the same α , several systems are intrinsically heterogeneous and non-stationary [19,20], which can lead to non-ergodicity (i.e., the non-equivalence of time and ensemble averages). Typically, the exponent α of the EA-MSD characterizes the physical properties of the systems (e.g., the trapping time distribution in CTRW or the time-dependence of diffusivity in SBM). However, in several non-ergodic systems, the TA-MSD shows a linear behavior with respect to the timelag in the the long time limit even when $\alpha \neq 1$ [4]; ...”

RESULTS

Comment R3.3 - *line 205 states that algorithms were grouped into three classes (ML over raw, ML over stat, classical state) but the table does not present them this way, Rather, the table defines a column for class (two choices), input (two choices), and L-specific (yes/no). The description should be harmonized to the table.*

Reply to R3.3 Following this comment (and also a suggestion by reviewer 1), we have restructured this section and added a reference to Table 1, where all teams are listed and classified according to the type of approach, input data, and length-specificity. The sections now reads:

“Among the participants, we could distinguish fifteen substantially different approaches (Table 1 and Supplementary Note 1). We classify the approaches based on three different criteria,

as detailed in Table 1. First, we group methods based on the type of approach used, whether involving machine-learning or classical statistics. A large majority of methods are based on machine-learning architectures, such as recurrent neural networks (RNN), convolutional neural networks (CNN), gradient boosting machines, graph neural networks, extreme learning machine (ELM), or sequence learners. Other methods are based on statistical approaches, such as Bayesian inference, temporal scaling, and random interval spectral ensemble (RISE). A second grouping involves the type of input data used. Some methods employed feature engineering using classical statistics as an input, whereas others were simply fed raw trajectories. A further classification is based on whether methods required a specific training or model for different (ranges of) trajectory lengths (length-specific) or not. Several methods could be directly used or easily adapted to run multiple tasks. ”

METHODS

Comment R3.4 - line 757 SNR was defined as $\sigma_D/\sigma_{\text{noise}}$ but I cannot find a definition of σ_D . It's probably somewhere but I'm not seeing it.

Reply to R3.4 We apologize, the definition has been added. The text now reads:

“The SNR was calculated as $\text{SNR} = \sigma_D/\sigma_{\text{noise}}$, where σ_D is the standard deviation of the distribution of displacements over unit time.”

Comment R3.5 - line 790: what is meant by the “amplitude” of the TA-MSD? Unless the motion is confined (which is not considered here), the curves grow without bound.

Reply to R3.5 We agree with the reviewer that this wording is confusing, therefore we have changed the text so that it now reads:

“the value of the TA-MSD for different trajectories at a given timelag ...”

Comment R3.6 - line 806: what is meant by “A mention apart...” This is likely an English usage problem, but I do not understand what the authors mean by this phrase.

Reply to R3.6 The text has been changed, it now reads:

“For the case of LW, a different kind of ergodicity breaking named ultraweak can be identified, where time and ensemble averages only differ by a constant factor.”

Comment R3.7 - in Algorithm 1, the sampling of the displacement appears to be a scalar. For higher dimensional motion, is it assumed the different directions are independent?

Reply to R3.7 For the sake of simplicity, the description of the algorithms is provided for the 1D case. The extension to 2D and 3D is provided later. We have further stressed this choice in the first paragraph of the Section “Theoretical models” that now reads:

“... as well as the description of the pseudocode used for simulations in 1D. Finally, we describe how to extend the algorithms to simulate the diffusion models in 2D and 3D, since for some models this is not simply obtained as the composition of motion along independent directions.”

Comment R3.8 - *In Algorithm 3, more details are needed on how sigma/gamma are generated. It simply says “generate randomly”. What underlying distribution is used?*

Reply to R3.8 We have modified the description of Algorithm 3 to clarify this point:

“ $\sigma \leftarrow$ uniform random number $\in (0, 3]$ ”

Comment R3.9 - *In Algorithm 4, I do not understand what the notation $(0, 1, \dots, T)^\alpha$ means.*

Reply to R3.9 We have modified the description of Algorithm 4 to clarify the notation:”

“ $\vec{\Delta x} \leftarrow (1^\alpha, 2^\alpha, \dots, T^\alpha) - (0^\alpha, \dots, (T - 1)^\alpha)$ ”

Comment R3.10 - *Only at line 871 was it clear to me that the previous material was all just about 1D. The model used does not allow for correlations between the different directions.*

Reply to R3.10 We have modified the first paragraph of the Section “Theoretical models” as explained in the comment about Algorithm 1 above. We now clarify that we first describe algorithm details for the 1D case, and then extend the description to the 2D and 3D cases.

Comment R3.11 - *Line 896: assigning a probability mass function to the set of models but then just taking the most likely? What if the approach is essentially tied between two? It seems odd to require a PMF but then assign an all-or-nothing score as if a point estimate was given. This is addressed by the ROC metric, I suppose.*

Reply to R3.11 Several different metrics can be used to evaluate the performance of classification methods, depending on the specific objective. For task 2 we used the F1 score because it evaluates the balance between the precision and recall of the classifier. For the calculation of this metrics, one only needs to provide the class of choice among the five models. However, for completeness, we wanted to calculate other metrics and compare them with the F1-score. For this, we chose the ROC/AUC that measures how well predictions are ranked. The calculation of the ROC/AUC requires methods to output a score for each class. Therefore we asked participants to provide the latter information to allow us calculating the ROC/AUC. For the calculation of the F1-score, we just considered the label of the class corresponding to the maximum score. The event of a perfect tie is very unlikely and, in fact, never happened. As shown in Supplementary Fig. 20, the two metrics resulted in similar rankings. The explanation is provided in “Methods. Additional metrics. Receiver operating characteristic (ROC) curve and area under the curve (AUC)”:

“The calculation of the F_1 -score assumes that a method outputs a discrete classifier (i.e., a unique choice for the diffusion model). However, many methods output continuous numbers associated to the probability of the input to belong to each class. Thus, these values assigned to

each model contain more information about the performance of the classifier. This information can be summarized by the ROC curve and the corresponding AUC. The ROC curve reports the true positive rate (or sensitivity) versus the false negative (one minus the specificity) for different levels of probability thresholds: if an input has a certain class probability above the threshold, it is considered to belong to such class. The AUC is given by the integral of the ROC curve and is equal to the probability that a classifier will rank a randomly chosen positive instance higher than a randomly chosen negative one. It thus provides a useful tool to compare the sensitivity and specificity of a given classifier.”

Comment R3.12 - Line 914: what is $\text{rank}_{\text{MAE}, \text{F1}, \text{RMSE}}$? The meaning of rank is undefined. Presumably it's the position in an ordered list based on performance in that metric but it is not clear.

Reply to R3.12 The reviewer is correct, the rank corresponds to the position in an ordered list based on metrics value. The definition has been added:

“... rank_{MAE} , rank_{F1} , and $\text{rank}_{\text{RMSE}}$ correspond to the position in an ordered list based on the value of the corresponding metrics.”

Comment R3.13 - Line 961: What is the definition of a “matching changepoint” Does it need to be an exact time point or is there a window?

Reply to R3.13 The section has been largely modified to provide a clearer explanation of the procedure:

“For the assessment of the changepoint localization error in T3, we followed two different evaluation approaches. For the challenge evaluation, we simply quantified the RMSE. Trajectories showing no changepoint were considered as having a dummy changepoint either at index 1 or 199. However, to get a better understanding of methods’ performance, we also considered an alternative analysis. For this, trajectories with ground truth and predicted changepoints within a distance $\epsilon = 20$ from the start/end points were considered as not having a changepoint. We thus considered four cases:

- predicted and ground-truth positions located at $\epsilon < t < L - \epsilon$, counted as true positives (TP);
- predicted and ground-truth positions located at $t \leq \epsilon$ or $t \geq L - \epsilon$, counted as true negatives (TN);
- the predicted position located at $\epsilon < t < L - \epsilon$ but the ground-truth located at either $t \leq \epsilon$ or $t \geq L - \epsilon$, counted as false positive (FP);
- the predicted position located at either $t \leq \epsilon$ or $t \geq L - \epsilon$. but the ground-truth located at $\epsilon < t < L - \epsilon$, counted as false negative (FN).

Based on this classification, we evaluated the recall (also known as sensitivity):

$$\text{recall} = \frac{\text{TP}}{\text{TP} + \text{FN}}; \quad (7)$$

the false positive rate:

$$\text{FPR} = \frac{\text{FP}}{\text{FP} + \text{TN}}; \quad (8)$$

and the Jaccard similarity coefficient (JSC) for binary classification:

$$\text{JSC} = \frac{\text{TP}}{\text{TP} + \text{FP} + \text{FN}}. \quad (9)$$

We also calculated the RMSE_{TP} , corresponding to the RMSE obtained only for prediction/ground-truth pairs classified as true positives.”

SUPPLEMENTAL 1

Comment R3.14 - *I realize it may not be feasible at this time, but I think have each time provide an optional (?) graphic for their approach would be helpful. This is especially true for ML methods where that visual can quickly and effectively convey the structure of the network that was used.*

Reply to R3.14 Our intention was to provide basic information about each method in the Supplementary file. A detailed description of each methods with clear and explanatory schemes can be found in the referenced articles. We have now stated it in the section “Participating teams and methods” (lines 216-218):

“Basic information about methods used by participating teams can be found in Supplementary Note 1. A detailed description of each methods can be found in the referenced articles.”

REVIEWERS' COMMENTS

Reviewer #1 (Remarks to the Author):

I thank the authors for the careful revision of their manuscript based on the reviewers' feedback. The authors have satisfactorily addressed my previous comments, and I think the manuscript is clearer and more complete now, especially the presentation and discussion around Task 3.

I have a couple of remaining comments:

(1) The manuscript states that 4 teams participated in T3 (p. 8). However, the figures and text then only show and discuss the results of 3 methods/teams. More confusingly, these 3 are sometimes referred to as the top 3 (e.g. in legend of Fig. 4a-c), and sometimes as "all methods" (e.g. at beginning of last paragraph on p.13 and in the legend of Fig. 4d, e). Which one is it? If there are truly 4 methods/teams, why not show them all? Omitting just one method seems unnecessary. Suppl. Fig. 5 also only shows the results of 3 teams.

(2) The new Suppl. Fig. 29 nicely shows the correlation in performance for shorter trajectories vs. all trajectories. It will be helpful to show in all panels a line indicating $y=x$, which would indicate completely equivalent performance. With such a line, one sees that while there is correlation in the performance, the performance for shorter trajectories is almost always a bit worse (not surprising). With this, I believe the authors should tone down the statement on p. 18 that "Top-ranking methods provide precise and unbiased predictions also for short and noisy trajectories." The authors should acknowledge that the performance is a bit worse for shorter trajectories and noisier trajectories (from Fig. 2e).

Reviewer #2 (Remarks to the Author):

The paper is worthy and timely and I strongly recommend its publication.

I still have an issue with the data analysis of the optical lattice experiment. Recall that fig. 5h shows clearly values of α less than unity, say $\alpha \simeq 0.8, 0.9$ namely your analysis makes the claim that the diffusion is anomalous. The microscopic theory predicts normal diffusion and the experiment was compared to this theory with no fitting parameters with great success (mentioned PRL of Dechant and Lutz).

My conclusion is that, in this case, the methods used by authors cannot provide insight, maybe because of the short available trajectories (as mentioned by authors).

Another option is that in the training session authors did not use the Langevin picture, which is the correct model. Indeed, as the authors write in their reply: the training might play a role.

Hence, we have here a classical case, machine learning can indeed distinguish between a dog and a cat, but when presented a donkey it yields non-sensible results. In particular, authors could state explicitly that possibly the fact that training did not use Langevin simulations, ended up with an estimation of α which is in conflict with the microscopical theory (note that this theory is based on Langevin not CTRW with $\alpha=1$, the latter is an approximation of the former).

I think a clearer discussion on the limitations of the methods is useful for a wide audience. Saying this differently, if the training has a negative role, what is it and what are the consequences?

Reviewer #3 (Remarks to the Author):

The authors have done an excellent job in responding to my concerns, addressing those that made sense and rebutting those that did not. I think this is a valuable paper and have only a few remaining comments that are mainly identifying remaining typos that I noticed when reading:

- Lines 177-180: I know this was in response to a different reviewer but I think the new text is overly negative. I suggest something like: While this choice limited the methods assessed to those provided by the community, it ensured that those algorithms were properly applied.

- line 218: a 'the' is missing. Should be "A detailed description of each of the methods..."

- lines 263-264: another missing 'the', now with a missing 'is'. Should be "...therefore, the result is indistinguishable..."

- line 375: another missing 'the'. Should be "...even when the odds of predicting..."

Point-by-point reply (II) to reviewers of *Objective comparison of methods to decode anomalous diffusion* by G. Muñoz et al.

Reply to Reviewer 1

Comment R1.1 *I thank the authors for the careful revision of their manuscript based on the reviewers' feedback. The authors have satisfactorily addressed my previous comments, and I think the manuscript is clearer and more complete now, especially the presentation and discussion around Task 3.*

Reply to R1.1 We thank the reviewer for the appreciation of our work.

Comment R1.2 *I have a couple of remaining comments:*

(1) The manuscript states that 4 teams participated in T3 (p. 8). However, the figures and text then only show and discuss the results of 3 methods/teams. More confusingly, these 3 are sometimes referred to as the top 3 (e.g. in legend of Fig. 4a-c), and sometimes as "all methods" (e.g. at beginning of last paragraph on p.13 and in the legend of Fig. 4d, e). Which one is it? If there are truly 4 methods/teams, why not show them all? Omitting just one method seems unnecessary. Suppl. Fig. 5 also only shows the results of 3 teams.

Reply to R1.2 We apologize for the confusing information. Technically, we had 4 teams participating to task 3, but one obtained rather poor results, comparable with random predictions. Therefore, the associated results were omitted from the discussion. We have now inserted this information in the manuscript (lines 219–220 of the version with highlighted changes):

"One of the methods participating to T3 had results comparable with random predictions and was thus excluded from the discussion of the results."

We now always refer to participants to task 3 as "all methods".

Comment R1.3 *(2) The new Suppl. Fig. 29 nicely shows the correlation in performance for shorter trajectories vs. all trajectories. It will be helpful to show in all panels a line indicating $y = x$, which would indicate completely equivalent performance. With such a line, one sees that while there is correlation in the performance, the performance for shorter trajectories is almost always a bit worse (not surprising). With this, I believe the authors should tone down the statement on p. 18 that "Top-ranking methods provide precise and unbiased predictions also for short and noisy trajectories" The authors should acknowledge that the performance is a bit worse for shorter trajectories and noisier trajectories (from Fig. 2e).*

Reply to R1.3 We thank the reviewer for the suggestion. We completely agree and modified the figures as suggested. To better explain our point we have also rephrased the text that now states (line 500 of the version with highlighted changes):

"Despite a little degradation of performance, top-ranking methods perform best also for short and noisy trajectories..."

Reply to Reviewer 2

Comment R2.1 *The paper is worthy and timely and I strongly recommend its publication.*

Reply to R2.1 We thank the reviewer for the positive assessment of our work.

Comment R2.2 *I still have an issue with the data analysis of the optical lattice experiment. Recall that fig. 5h shows clearly values of alpha less than unity, say alpha \simeq 0.8, 0.9 namely your analysis makes the claim that the diffusion is anomalous. The microscopic theory predicts normal diffusion and the experiment was compared to this theory with no fitting parameters with great success (mentioned PRL of Dechant and Lutz).*

My conclusion is that, in this case, the methods used by authors cannot provide insight, maybe because of the short available trajectories (as mentioned by authors). Another option is that in the training session authors did not use the Langevin picture, which is the correct model. Indeed, as the authors write in their reply: the training might play a role. Hence, we have here a classical case, machine learning can indeed distinguish between a dog and a cat, but when presented a donkey it yields non-sensible results. In particular, authors could state explicitly that possibly the fact that training did not use Langevin simulations, ended up with an estimation of alpha which is in conflict with the microscopical theory (note that this theory is based on Langevin not CTRW with $\alpha = 1$, the latter is an approximation of the former). I think a clearer discussion on the limitations of the methods is useful for a wide audience. Saying this differently, if the training has a negative role, what is it and what are the consequences?

Reply to R2.2 We have now made explicit the possible effect caused by the lack of Langevin simulations in the training dataset (lines 478–479 of the version with highlighted changes):

“Predictions might also suffer from the lack of training data based on the microscopic model of Ref.[56], of which CTRW with $\alpha = 1$ is an approximation.”

Reply to Reviewer 3

Comment R3.1 *The authors have done an excellent job in responding to my concerns, addressing those that made sense and rebutting those that did not. I think this is a valuable paper and have only a few remaining comments that are mainly identifying remaining typos that I noticed when reading:*

- Lines 177-180: I know this was in response to a different reviewer but I think the new text is overly negative. I suggest something like: While this choice limited the methods assessed to those provided by the community, it ensured that those algorithms were properly applied.

- line 218: a 'the' is missing. Should be "A detailed description of each of the methods..."

- lines 263-264: another missing 'the', now with a missing 'is'. Should be "...therefore, the result is indistinguishable..."

- line 375: another missing 'the'. Should be "...even when the odds of predicting..."

Reply to R3.1 We thank the reviewer for the careful reading of our manuscript. We have corrected the text as suggested.